## REVIEW ARTICLE

# Biology and therapeutic potential of mesenchymal stem cell extracellular vesicles in axial spondyloarthritis

Fataneh Tavasolian [1] & Robert D. Inman [1,2,3✉]

Axial spondyloarthritis (AxSpA) is a chronic, inflammatory, autoimmune disease that predominantly affects the joints of the spine, causes chronic pain, and, in advanced stages, may result in spinal fusion. Recent developments in understanding the immunomodulatory and tissue-differentiating properties of mesenchymal stem cell (MSC) therapy have raised the possibility of applying such treatment to AxSpA. The therapeutic effectiveness of MSCs has been shown in numerous studies spanning a range of diseases. Several studies have been conducted examining acellular therapy based on MSC secretome. Extracellular vesicles (EVs) generated by MSCs have been proven to reproduce the impact of MSCs on target cells. These EVs are associated with immunological regulation, tissue remodeling, and cellular homeostasis. EVs' biological effects rely on their cargo, with microRNAs (miRNAs) integrated into EVs playing a particularly important role in gene expression regulation. In this article, we will discuss the impact of MSCs and EVs generated by MSCs on target cells and how these may be used as unique treatment strategies for AxSpA.

AxSpA is an autoimmune disease resulting from a complicated interplay between genetics and the environment[1]. Despite breakthroughs over the last several decades, the etiology of AxSpA remains unknown. Genetic background, immunological response, microbial infection, and endocrine dysregulation are some of the factors linked to the development of AxSpA in previous studies[2]. HLA-B27 has been associated with AxSpA in numerous populations worldwide, but the mechanisms remain unclear[3]. HLA-B27 is present in 90–95% of AxSpA patients, although only 1–2% of HLA-B27-positive individuals will develop AxSpA[4]. HLA-B27 is found on the cell surface in both heterodimeric and homodimeric forms and intracellular and exosomal MHC-I dimers[5,6]. In recent years, HLA non-B27 and non-HLA genes have been identified in AxSpA as a result of genome-wide association studies (GWAS) and other technologies. ERAP1 and IL23R provided early evidence for an important role for numerous non-MHC genes associated with AxSpA[7,8].

### Factors affecting the immune system and microbes

AxSpA has been linked to a variety of autoimmune diseases, such as IBD, anterior uveitis, and psoriasis, implicating a common genetic and immunological basis. A recent study indicates that mononuclear cells of AxSpA patients generate IL-23 and IL-17 in response to inflammasome activation[9]. Inflammasomes are key elements of the innate immune system that organize antimicrobial host defenses and control the inflammatory response[10]. Also, the researchers demonstrate that synovial fluid (SF) in AxSpA patients is enriched with CD8+ CTLs with distinctive integrin expression patterns that suggest gut-joint trafficking[11,12]. Microbial infection

[1] Spondylitis Program, Division of Rheumatology, Schroeder Arthritis Institute, University Health Network, Toronto, Ontario, Canada. [2] Krembil Research Institute, University Health Network, Toronto, Ontario, Canada. [3] Departments of Medicine and Immunology, University of Toronto, Toronto, Ontario, Canada. ✉email: Robert.Inman@uhn.ca

may act as a catalyst for the development of AxSpA by activating the host's innate immune system. HLA-B27 transgenic rats do not develop AxSpA symptoms in a germ-free environment, but this changed when commensal bacteria were administered to the germ-free animals, revealing possible links between HLA-B27 and the microbiome. The gut microbiome is critical for gut homeostasis, and its disruption has major implications for immune control and the progression of AxSpA disease[8,13]. Although it is now accepted that individuals with AxSpA have an altered gut microbiome, no consistent and uniform pathogen profile has been identified. The gut microbiome is thought to have a role in AxSpA through interacting with genetic, immune-mediated, and microbial metabolic disorders[12].

### Inflammation and new bone formation in AxSpA

In AxSpA, axial and peripheral joints often exhibit chronic inflammation, and in the case of the spine, can be accompanied by the formation of new bone. The most often affected sites of inflammation are the spinal vertebral corners, sacroiliac joints, sternal rib junction, iliac crest, ischial tuberosity, Achilles tendon, and plantar fascia[14,15]. Despite the fact that magnetic resonance imaging may identify inflammation before the development of radiographic abnormalities and that early pharmaceutical intervention can reduce inflammation, successfully preventing the formation of new bone remains a challenge. Identifying components responsible for new bone formation might pave the way for the creation of novel therapeutics for AxSpA. It is believed that genetics, immune cell interactions, inflammatory cytokines, and anabolic signaling pathways influence inflammation and subsequent bone formation[16].

Several studies have demonstrated osteogenic differentiation of MSCs from AxSpA patients[3,17–20]. These findings suggesting pathogenic potential, together with the therapeutic efficacy of MSCs infusion in both preclinical and clinical studies, suggest that MSCs may play a dual role in the development and treatment of AxSpA. This review summarizes the MSCs' cell biology and their therapeutic advancement, analyzes the many roles of MSCs in the development and treatment of AxSpA, and presents EVs derived from MSCs as a possible innovative therapy for AxSpA.

### Mesenchymal stem cell point of view: cell biology to therapeutic advancement

While much research on cell-based therapies has focused on regenerative medicine, there remains the hope of employing cell therapies as new, alternative treatments for various diseases. Human multipotent MSCs are now being studied for their possible application in therapy for several diseases. Due to their capacity for differentiation, immunomodulation, and paracrine factor release, MSCs have attracted considerable attention as candidates for cellular therapies[21]. MSCs are nonhematopoietic cell progenitors first isolated in bone marrow (BM) but now recognized to be present in a variety of other tissues. They possess the capacity for self-renewal and display limited differentiation[22–24] (Fig. 1). MSCs may be found in the stroma of all adult organs, although they are most often found in perivascular areas, where they contribute to tissue homeostasis, monitoring, repair, and remodeling. The niche of the human BM consists of nonhematopoietic cells that provide physical support for hematopoietic stem progenitor cells (HSPCs) and maintain their homeostasis. MSCs are essential components of the BM

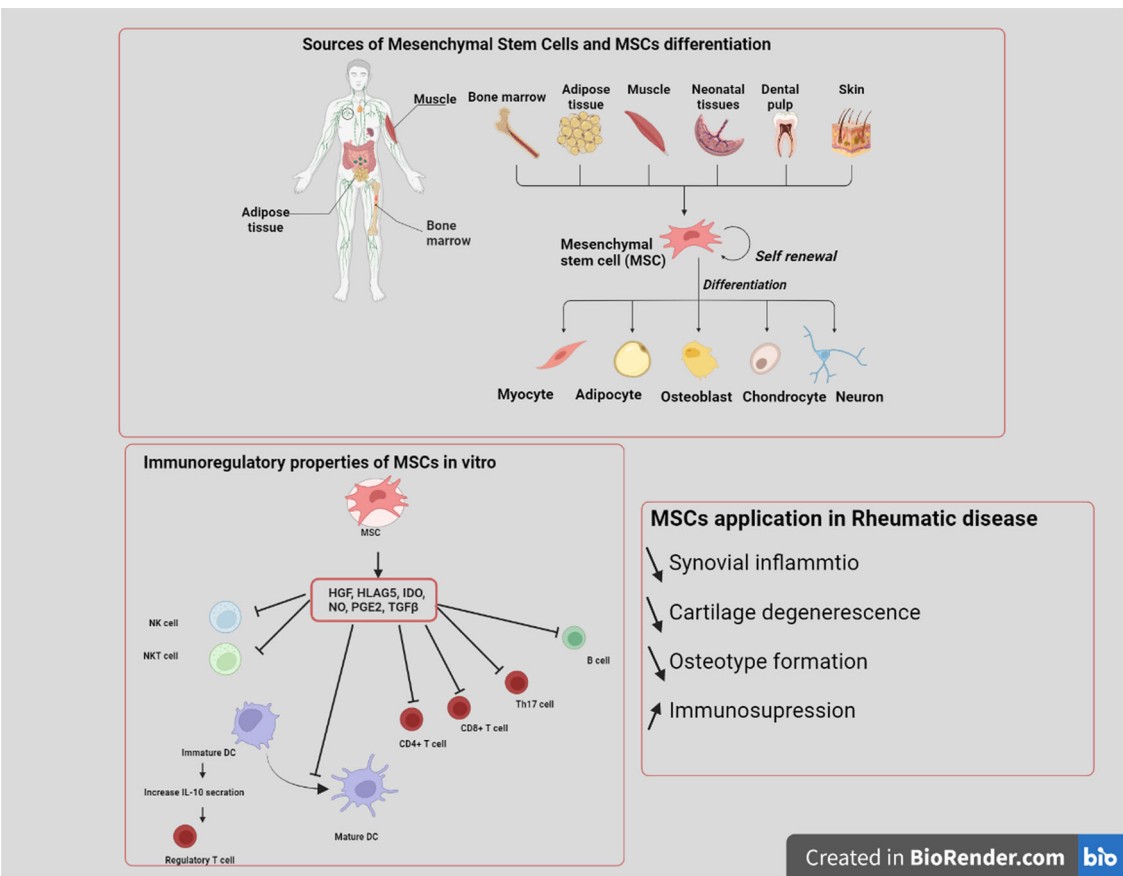

**Fig. 1 The properties and applications of mesenchymal stem cells.** MSCs are composed of multipotent stem/progenitor cells. Under certain conditions in vitro and in vivo, MSCs may differentiate into distinct lineages. MSCs exhibit significant anti-inflammatory and immunomodulating properties.

microenvironment, where they provide newly formed osteoblasts for bone tissue regeneration and tightly regulate the fate of HSPCs through direct interaction and the secretion of soluble factors, thus playing a crucial role in the development and differentiation of the hematopoietic system[25–27]. Researchers have examined MSCs utilizing diverse separation and development procedures to characterize the cells. Comparing and contrasting studies becomes more challenging. In order to address this issue, the Mesenchymal and Tissue Stem Cell Committee of the International Society for Cellular Therapy has established basic criteria for characterizing human MSCs. First, the MSCs must adhere to plastic when grown under standard circumstances. MSCs must also express the surface molecules CD105, CD73, and CD90, but not CD45, CD34, CD14, CD11b, CD79, CD19, or HLA-DR. Third, MSCs must be capable of differentiating into osteoblasts, adipocytes, and chondroblasts in vitro. Despite the likelihood that these criteria may need to be modified in the future, it is anticipated that this basic set of standard criteria will result in a more consistent categorization of MSCs and will simplify the sharing of data among researchers[28,29].

## Modulation of the immune system by MSCs

Because of their anti-inflammatory, immunomodulatory, and regenerative capabilities, MSCs have been studied for use in cell-based therapeutics[30]. Paracrine and cell-to-cell contact pathways mediate these responses. Paracrine effects are mediated by the MSC secretome, which comprises cytokines, chemokines, and microvesicles/exosomes that transport proteins or miRNAs to target cells[28]. Concurrent with early MSC/hematopoietic stem cell transplant clinical studies, in vitro studies tested allo-MSCs in mixed donor lymphocyte reactions and found that the MSCs inhibited lymphocyte proliferation and did not induce apoptosis of T cells; rather, T cells responded to subsequent lymphocyte challenge when the MSCs were removed[31,32]. Numerous additional research has validated similar results. For cell–cell interaction, it was shown that MSCs ordinarily display major histocompatibility complex (MHC) class I antigens on their surface but not class II antigens; nevertheless, inflammatory factors upregulate class II antigens. The exhaustive search for soluble factors secreted by MSCs that cause them to be immunomodulatory uncovered multiple factors that limit immune cell responses, including transforming growth factor β, hepatocyte growth factor, prostaglandin E2, interleukin-10, interleukin-1 receptor antagonist, interleukin-6, human leukocyte antigen-G, leukocyte inhibitory factor[33–39]. MSCs also biased maturing immune cell populations, resulting in an increase in regulatory T cells (Treg), anti-inflammatory TH2 cells, and dendritic cells type 2, and a decrease in pro-inflammatory TH1 cells, dendritic cells type 1, and NK cells. MSCs also encouraged M1 macrophages to transform into the anti-inflammatory M2 form and inhibited B cell IgG production[40]. While several of these discovered components have been utilized singly to suppress immunological responses, the MSCs generate a more comprehensive immune modulation as a result of numerous factors functioning in concert. MSCs are hypoimmunogenic under homeostatic settings as a consequence, making them appropriate for allogeneic transplantation. Chemokine receptors, matrix metalloproteinases (MMPs), and adhesion molecules are all abundantly expressed in MSCs, which might contribute to their migration to sites of inflammation[41,42].

## Do MSCs create more problems than they solve in AxSpA?

**Bone remodeling in AxSpA.** The coupling activity of osteoblasts (OBs) and osteoclasts (OCs), under normal physiological conditions, maintains the dynamic equilibrium of bone formation and bone resorption[43–45]. In contrast to OCs, which orchestrate bone loss, OBs generate an organic matrix and aid mineralization. The interactions between cells tightly regulate this bone remodeling process. For bones to maintain their mechanical integrity and strength, synthesis and absorption must be appropriately balanced[46]. OC and OB activities, however, become uncoupled under inflammatory situations, leading to excessive bone resorption or formation[47]. The alteration of bone exist in AxSpA by coexisting of osteolysis and osteogenesis. Early inflammatory lesions of localized hyperemia and edema are associated with OB activity, and bone marrow edema is often significant when radiographic signs of joint injury have not yet appeared[48,49]. As a result, more OCs may be the prime culprits in radiographic joint damage during acute inflammation in AxSpA.

Over time, chronic inflammation induces an anabolic skeletal reaction, with new cortical bone synthesis at sites of inflammation. This can be associated with excessive trabecular bone resorption, and the trabecular bone loss is frequently found to be connected to the formation of new bone at the enthesis sites[49,50]. MSCs have a higher capacity for osteogenic differentiation at this time, and OBs form more ossification foci in the subchondral granulation tissue, which may precede the formation of marginal syndesmophytes. Therefore, the degree of local bone inflammation in the spine is thought to be the precursor of spinal radiographic damage in AxSpA[3,51].

The complex interaction of cytokines and signaling pathways released by various immune cells on bone cell activity and bone mass has been increasingly clarified. The immune system modulates distinct bone cell types differently at different stages of the disease[44,51,52].

## MSCs demonstrate enhanced osteoblast differentiation in AxSpA

MSCs are essential for maintaining bone homeostasis as they may undergo trilineage differentiation into OBs, chondroblasts, and adipoblasts to take part in bone remodeling[53]. Due to their immunomodulation abilities, including their capacity for self-renewal and multipotent differentiation, MSCs may play a role in AxSpA[54,55]. MSC osteogenic development is controlled by several intracellular signaling networks, including the BMP/Smad pathway, the WNT/catenin pathway, and the MAPK system. These signaling pathways also play a role in the pathological osteogenesis associated with AxSpA. The reason for the increased osteogenic differentiation potential of MSCs in AxSpA has been sought[56]. BM-MSCs play a vital role in healthy joints by preserving bone homeostasis and repairing damaged tissues. However, selective RANKL expression in MSCs may contribute to joint inflammation in an inflammatory environment. In AxSpA, this could result in the binding of RANKL to RANK in inflammatory MSCs, thereby contributing to reverse signaling in osteoblasts and promotion of osteoblast differentiation (Fig. 2)[57–60].

BM-MSCs from AxSpA patients have a greater intrinsic ability for osteogenic growth than BM-MSCs from healthy donors[19]. An imbalance between enhanced BMP2 and decreased Noggin secretion was connected to AxSpA-MSC osteogenic differentiation, according to studies examining the osteogenic differentiation capability of sternal BM-MSCs from AxSpA[3]. In AxSpA patients, BMP2 expression was considerably greater in BM-MSCs from ossifying entheses. Increased osteogenic differentiation is a consequence of BMP2 overexpression[3]. MCP1 is another factor that MSCs generated more during abnormal osteogenic differentiation in AxSpA and induces monocyte dysfunction. Therefore, aberrant osteogenesis might result in AxSpA inflammation[20]. Additionally, it is recognized that the

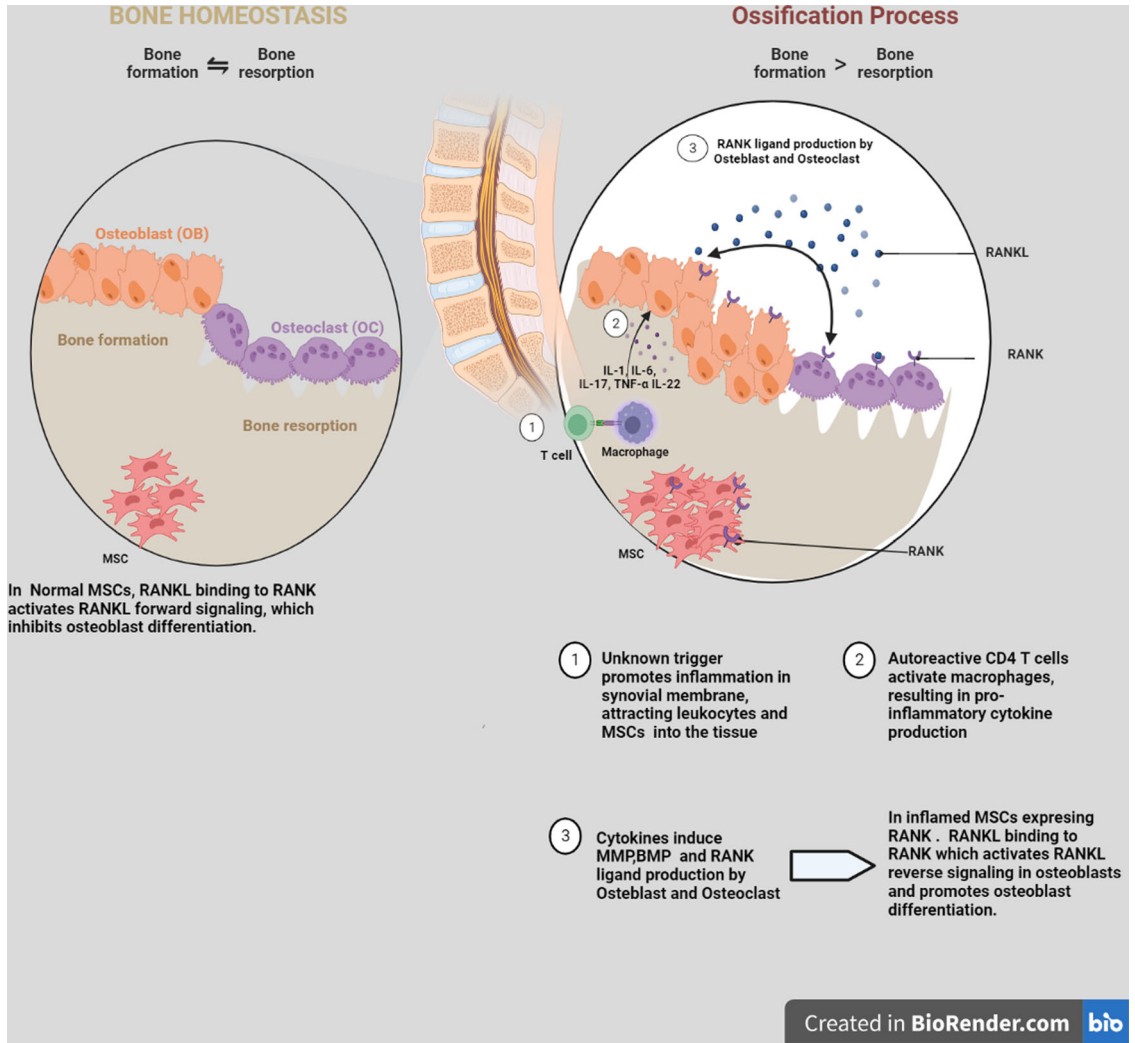

**Fig. 2 The RANKL-RANK signaling functioning model that is suggested controls osteoblast differentiation and bone formation in inflammatory conditions in AxSpA.** BM-MSCs provide a crucial job in normal joints by maintaining bone homeostasis and repairing damaged lesions due to their distinct normal environment activities. However, selective RANKL expression in MSCs may contribute to joint inflammation in an inflammatory environment. MSCs are thus candidate target cells for TNF in these disorders. Different cells secreting IL-22 in entheses provide an additional option for the involvement of BM-MSCs in AxSpA. The majority of studies evaluating the function of BM-MSCs in the pathophysiology of AxSpA have focused on their engagement in the ossification of entheses, which is characteristic of persistent AxSpA. In inflammatory MSCs expressing RANK, RANKL binding to RANK stimulates RANKL to reverse signaling in osteoblasts and promotes osteoblast differentiation[57].

immunomodulatory capacity of MSCs from AxSpA patients is decreased, possibly due to an imbalance between CCR4+ CCR6+ Th/Treg cells[61]. Different cytokine concentrations affect MSC regulation at various levels[62]. For instance, IL-17 is elevated in AS patients and suppresses DKK-1 expression while promoting osteoblastic activity[63]. Dickkopf-1 (Dkk-1) is an essential regulator of bone remodeling in spondyloarthropathies. The expression of IL-17A on neutrophil extracellular traps stimulates the osteogenic potential of MSCs[18].

While low levels of IL-17A promote polarization of TLR4+ MSC and inhibit osteogenic differentiation via the JAK2/STAT3 pathway, high levels of IL-17A promote TLR3+ MSC polarization and enhance osteogenic differentiation via the Wnt10b/Runx2 pathway[64]. Control of MSC apoptosis is also a key factor, and MSCs from AxSpA patients exhibit greater levels of apoptosis than healthy MSCs[65]. This is likely because MSCs elicit effector T cells by secreting chemokines that either mediate direct immunoregulation or cause Fas/FASL–induced apoptosis[65,66]. TNF-related apoptosis-inducing ligand receptor 2 (TRAIL-R2) is

expressed at higher levels in MSCs from AxSpA patients than in healthy MSCs, rendering them more vulnerable to TNF/CHX-induced apoptosis[52,67]. MSCs from AxSpA patients were shown to elicit TNF-mediated inflammatory responses and higher osteogenic differentiation[68]. In active AxSpA, the frequency of Treg and Foxp3+ cells was reduced, whereas the frequency of CCR4+ CCR6+ Th cells rose, indicating that those BM-MSCs had a poorer immunomodulatory potential[61]. MSCs from patients with AxSpA show lower immunoregulatory function[20,61,68,69].

## Therapy using external MSCs in AxSpA

External MSC implantations have been proven to have positive and protective effects on AxSpA diseases in both preclinical and clinical investigations. These MSCs are amplified in vitro from either autologous or allogeneic sources. MSCs may be directly injected into an inflammatory joint[70,71], and if this is not feasible, cells may be delivered by systemic injection, in which case external MSCs with homing ability may migrate to inflamed sites.

How MSCs from internal and exterior sources act differently throughout the formation of AxSpA poses a challenge to resolve. The key to answering this question may lie in the milieu that the internal MSCs encountered when localized in damaged tissue and that the external MSCs directly encountered during circulation or in synovial tissues following cell injection.

In the presence or absence of IL-23, enthesitis in AxSpA has a high number of immune cells that produce type 3 immunity-related cytokines (IL-17, IL-22, and GM-CSF)[72–74]. In concert with other potential risk factors, such as male gender, HLA-B27 status, and mechanical loading stress, these cytokines initiate and sustain inflammation in spinal entheses, resulting in the induction of new bone formation[75,76]. MIF and TNF are likewise released largely by myeloid cells and cause osteoproliferative alterations[77]. In response to new bone formation-initiating stimuli, osteochondral progenitor cells, such as MSCs or periosteal cells, differentiate into osteoblasts or chondrocytes to produce new bone by intramembranous or endochondral ossification, respectively. During differentiation, crucial anabolic molecules and signaling pathways are active, including BMPs, RANKL, and Wnt. These findings demonstrate that MSCs are exposed to a diverse microenvironment, and the diversity of environmental stimuli may result in a wide range of cellular responses[16,77].

## Clinical trials of MSCs in AxSpA

Many clinical trials examining MSC transplantation in rheumatic diseases are now underway, including phase I/II studies in AxSpA to determine the safety and therapeutic advantages of MSCs transplantation[40]. MSCs transplantation has been examined as a treatment option for AxSpA patients. The number of Treg cells in AxSpA patients has been shown to be decreased. MSCs may limit Th17 cell production by prompting T cells to develop into the Treg phenotype, decreasing the number of Th17 cells[78–80]. In a phase 1 clinical trial, human umbilical cord-derived MSCs (hUC-MSCs) were administered intravenously and repeated after 3 months, in combination with DMARDs (NCT01420432). In another trial, patients with AxSpA received infusions of human MSCs and Non-steroidal anti-inflammatory drugs (NSAIDs) (NCT01709656). A phase 2 clinical study (NCT02809781) is now underway to assess the use of human bone marrow-derived MSCs in AxSpA patients, as well as a phase I/II clinical trial (ChiCTR-TRC-11001417) to determine the safety of MSC treatment in AxSpA patients[70]. At this time, there is little consensus on their effectiveness. A 20-week clinical trial using allogenic MSCs administered intravenously was done with AxSpA patients who had failed to respond to NSAIDs. The absence of a control group limits the generalizability of the study[81]. A meta-analysis of randomized controlled trials concluded that 6 months of MSC treatment for AxSpA may reduce erythrocyte sedimentation rate, intercellular adhesion molecules, and serum TNF[82]. In addition, consideration should be given to the effect of MSC on the differentiation of innate lymphoid cells (ILCs), which are necessary to sustain tissue homeostasis and bridges between the innate and adaptive immune systems, and may aid in the development of ILC-based therapies for inflammatory disorders[83,84].

## Limitations of MSC-based therapy for AxSpA

Because it is challenging to collect MSCs from entheseal BM, most investigations on BM-MSCs from AxSpA patients have employed BM-MSCs from distant regions (such as the sternum) or produced pluripotent stem cells (such as dermal fibroblasts)[15,20,85]. Numerous cell-delivery strategies are ineffective, with many studies demonstrating that only a tiny fraction of injected cells stay at the injection site days after transplantation[86]. Transplanted MSCs have a limited period of viability in recipients

after undergoing apoptosis in the host's circulation or engrafted tissues[87]. Although the clinical trial design is receiving attention, the specific equipment and techniques used to implant the cells locally have lately been profiled more[88]. The remaining challenges include prices and potential adverse effects, which might lead to preclinical and clinical testing inconsistencies. Differential cell behavior, dose and distribution accuracy, and cell retention and survival after injection are only a few hurdles that must be overcome before meaningful translation can occur. The success of injectable cell transplantation depends on accurate measurement of post-injection cellular health and the development of consistent delivery mechanisms[89]. Consequently, prospective controlled trials are considered necessary to measure MSC-based therapy and determine its potential efficacy, specifically in treating AxSpA[90].

Nonetheless, as of January 2018, there was no FDA-approved medication for use in the United States[91–94]. A crucial obstacle is guaranteeing that the MSCs, when supplied to patients, will execute the desired targeted function. MSCs are very sensitive to their surroundings. In a lab-based production method, MSCs are exposed to an environment differing from conditions in vivo, raising the possibility of altering their response to growth factors and produce MSC preparations introducing unexpected and unwanted variability. Additionally, the behavior of the cells may change after being injected into a patient. Cell and nuclear morphology may serve as possible distinguishing characteristics of MSC potency[95,96]. The influence of morphology-directed stem cell lineage determination has been shown in both 2D and 3D and may serve as an early signal of osteogenic differentiation for MSCs[96–99]. It has also been established that the size of MSCs increases with passage and donor age; hence it is feasible that underlying morphological distinctions in MSC populations might explain or predict their variability in potency[100–102]. Like cell morphology, nuclear morphology has been recognized as a predictor of stem cell activity and a phenotypic indicator of epigenetic and transcriptional cellular activities[103].

## MSC-derived EVs: a novel cell-free therapy

**Extracellular vesicles**. Extracellular vesicles (EVs) are tiny vesicles generated by almost all cell types, characterized by a phospholipid bilayer and harboring a wide array of proteins, mRNAs, and miRNAs. Exosomes (diameter less than 150 nm) are formed in the endosomal compartment in so-called multivesicular bodies, and microvesicles, or microparticles (diameters range from 150 to 1000 nm), are released by plasma membrane budding[104]. The International Society of Extracellular Vesicles has published basic criteria to characterize EVs, including shape, the process of cellular release, and biochemical characteristics[105–107]. The therapeutic effectiveness of MSC-derived EVs (MSC-EVs) has been described in several disease models, including myocardial infarction and reperfusion damage, liver and kidney injury, hind limb ischemia, and inflammatory illnesses. Although there is considerable interest in MSC-EVs for the therapy of several illnesses, little is known about their precise function[108,109].

**Exosomes**. Exosomes are the most well-studied EV subclass. Exosome membranes are distinguished from endosomes by the presence of lipid rafts, which are involved in the fusion and release of intraluminal vesicles (ILV) and multivesicular bodies (MVB). MVB attaches to the plasma membrane, and exosomes are released. Membrane fusion, endocytosis, or cell type-specific phagocytosis may then be used by other cells to pick up exosomes[110]. The ability of exosomes to carry microRNAs, lipids, and proteins through tissue and biological barriers makes them promising as therapeutic vehicles (Fig. 3)[111]. The emerging

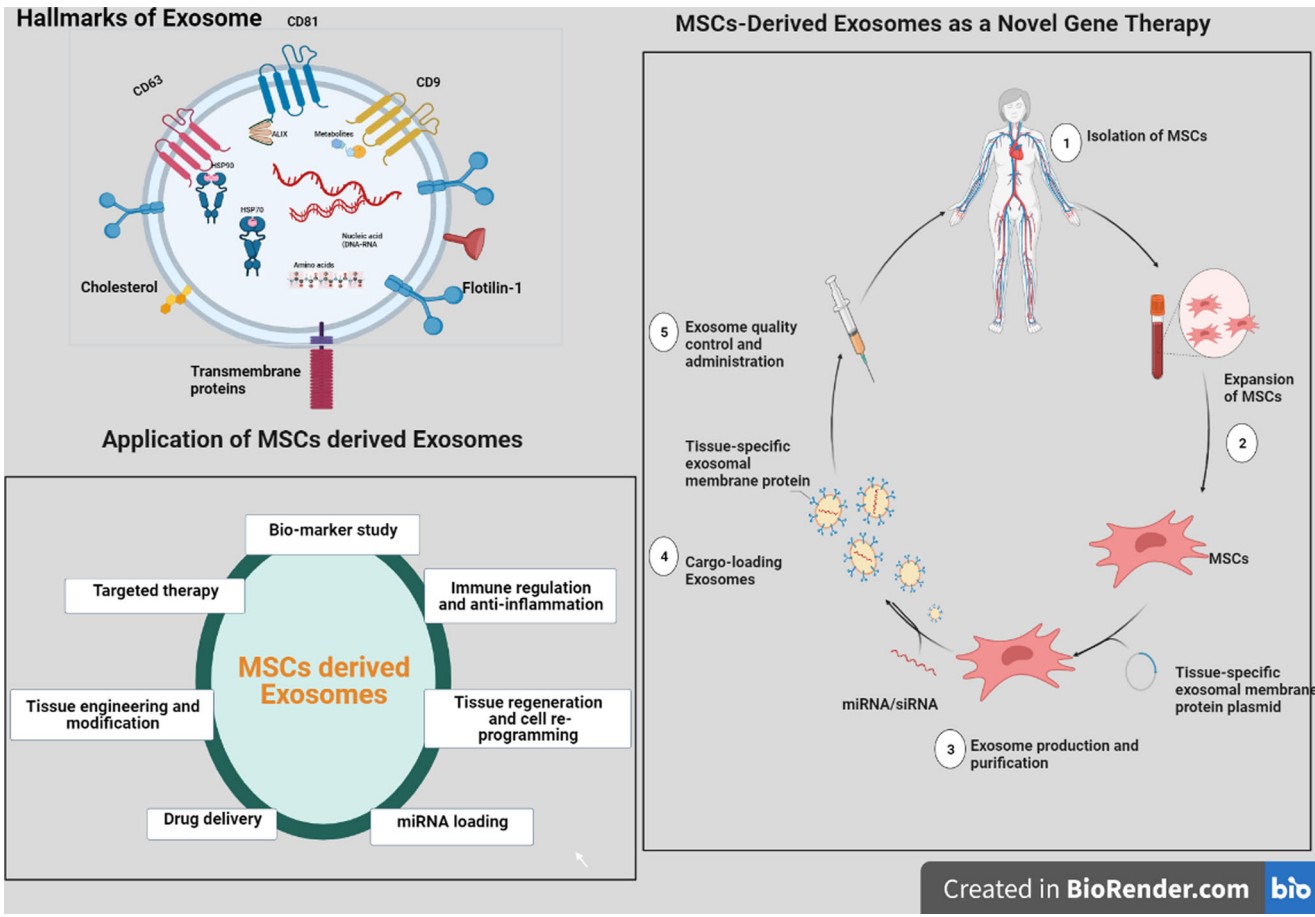

**Fig. 3 Exosome characterization, isolation from MSC, and application as novel gene therapy.** Exosomes are cell-secreted nanoparticles (30–150 nm in size) containing various biological components, including nucleic acids, proteins, and lipids, which play crucial roles in intercellular communication. As carriers, exosomes offer promise as enhanced platforms for targeted gene delivery due to their unique features, including intrinsic stability, minimal immunogenicity, and exceptional tissue/cell penetration potential. Targeted delivery raises the local concentration of therapeutics while minimizing negative effects.

consensus that exosomes operate as a mode of communication between and among cells and tissues is an appealing concept that could change the current understanding of disease pathogenesis[112]. It has been demonstrated that the protein profiles of serum-derived exosomes differed between AxSpA patients and healthy subjects. In a functional analysis, the differentially expressed proteins may contribute to alteration in immune responses. Differentially expressed proteins have been discovered in AxSpA serum-derived exosomes, which may provide new insights into the pathophysiology of AxSpA and could lead to the discovery of novel biomarkers for the disease[113–116].

**Microvesicles**. When a cell is stimulated or undergoes apoptosis, microvesicles (MV) are released by the outward budding and fission of the plasma membrane, while exosomes are produced via the inward budding of the limiting membrane of early endosomes[117]. MV exhibit excellent biocompatibility, minimal immunogenicity, and targeting, and may be employed as drug carriers. The use of microvesicles produced from tumor cells to transport chemotherapy medications has been found to improve cancer treatment outcomes with few unwanted effects[117,118].

**Therapeutics based on MSC-derived EVs**. Gnecchi et al. investigated myocardial regeneration and discovered that the conditioned media of MSCs could aid in vivo myocardial

regeneration[119]. While conditioned media has traditionally been regarded as a source of cytokines and growth factors that support regeneration or lineage-specific differentiation, Lai et al. demonstrated that the exosomes of MSC conditioned media positively influenced cardiac tissue regeneration and repair[120]. Since then, there has been a growing interest in using MSC-exosomes as a cell-free alternative to MSCs to direct tissue regeneration and tissue engineering[24]. Exosomes may exhibit MSC-assigned functions that regulate the proliferation, differentiation, migration, and apoptosis of diverse target cells and their functions. Exosomes derived from MSCs may offer comparable benefits, opportunities, and challenges in the context of bone/cartilage regeneration[121].

MSC-exosomes contain several microRNAs that encode signaling pathway regulators involved in repair and regeneration (e.g., ERK, SMAD)[122]. ExoCarta is a database of exosomes that contains details and statistics about exosomal cargo, such as identified miRNAs[123]. Vesiclepedia is a second freely accessible source for EVs[124].

Exosome miRNA content is characteristic of the parental cell and suggests that exosome miRNA content is selective and appears to be specific to the exosome-derived cell type and cell condition (e.g., hypoxia, inflammation). Additionally, it is feasible to direct miRNAs into exosomes using an EXO-motif[125]. Thus, miRNAs are amenable to therapeutic modification of cellular functions. The specificity of exosome miRNA cargo may also be

**Table 1 MSCs and MSC-exosomes for therapeutic applications: benefits and drawbacks.**

|  | Positive aspects | Negative aspects |
|---|---|---|
| MSCs | Simple to isolate and collect | Probability of transmitting infections |
|  | Highly prolific | Concerns over the associated regenerative process |
|  | Multilineage differentiation |  |
|  | Limited likelihood of immunological issues |  |
|  | Cumulated experimental and clinical outcomes |  |
| MSC-exosomes | Effectiveness via particular proteins in the exosome membranes and natural homing capability | A minimal isolation procedure is indicated |
|  | Low chance of teratoma development | No controlled production procedures |
|  | Excellent medication delivery system for both hydrophobic and hydrophilic substances | Quick elimination from the bloodstream upon injection (in vivo) |
|  | Unaffected by freezing and thawing (compared with cells) | Challenges in isolating and purifying exosomes containing certain bioactive compounds |
|  | Paracrine function | Deficiency of methods and tools to precisely characterize the chemical and physical characteristics of exosomes |
|  |  | Minimal and restricted investigations on exosome-based treatments |
|  |  | Probability of transmitting infections |

diagnostic. MiRNAs in EVs are also shielded from RNAse destruction, and their integrins and opsonins allow for selective delivery of their internal content[126–129]. The advantages and disadvantages of utilizing MSCs and MSC-derived EVs are summarized in Table 1. While less prevalent, lncRNAs may be essential as constituent exosomal cargo. lncRNA regulation of cell function during genetic reprogramming suggests that exosomal lncRNA may also be involved in exosome-mediated alterations in target cell function[130]. The exosome cargo appears to be a specifically encapsulated combination of proteins and RNAs that presents valuable new opportunities for diagnostics and therapies.

**Exosomes derived from MSCs and cartilage regeneration**. Exosomes from MSCs and chondrocytes have an effect on cartilage regeneration and repair. Exosomes isolated from human MSCs (derived from HuES9 human embryonic stem cells) induced cartilage and subchondral bone repair in rodents with osteochondral defects comparable to unoperated control sites after 12 weeks[131]. Exosomes represent a beneficial cell-free strategy for utilizing human embryonic MSCs for cartilage repair, as suggested by the researchers, who noted that many components of the MSC exosome were necessary for effective tissue regeneration[132]. Reviewing the potential use of exosomes for inducing chondrogenic differentiation and upregulating chondrogenic transcription factors[132,133]. In addition, the selective modification of the local population of regenerative M2 macrophages as opposed to pro-inflammatory M1 macrophages demonstrated that MSC exosome-mediated cartilage repair involved specific immune modulation[131]. Recent research has demonstrated the ability to tailor exosome-mediated regenerative therapies to specific clinical conditions. To directly increase the proliferation and migration of targeted chondrocytes without inhibiting extracellular matrix (ECM) protein synthesis, they engineered human synovial MSC-exosomes to contain elevated levels of the miR-140-5p. Exosomes from MSCs may modulate regeneration by exerting specific influences on distinct cells in the local environment[134].

**Exosomes derived from MSCs and bone regeneration**. Exosomes affect MSC osteoinduction, according to multiple studies. In vitro osteoblastic differentiation of mesenchymal stem cells can be stimulated by exosomes from osteoblastic cells, dendritic cells, and monocytes in cell culture[135–137]. Exosomes from MSCs and MSCs undergoing osteoblastic differentiation stimulate osteoblastic

differentiation. In addition, chondrocyte-derived exosomes directed the experimental development of subcutaneous tissues resembling cartilage. Antiangiogenic factors of chondrogenic exosomes were hypothesized to be responsible for maintaining a favorable niche for chondrogenesis. Numerous exosomes possess immunomodulatory properties indispensable for cartilage regeneration[132,138]. This was confirmed by a recent study on the role of M2 macrophage polarization in the resolution of osteochondral defects following MSC exosome therapy. In this study, cartilage repair and regeneration were accompanied by enhanced cell proliferation, decreased apoptosis, and increased matrix synthesis[131]. According to these studies of cartilage restoration, exosomes are capable of directing tissue-specific regeneration by targeting multiple biological and cellular processes.

In search of the molecular mechanisms of MSC-derived EV-mediated regenerative effects on bone and cartilage-forming cells, several components of the EV cargo as well as the targeted molecules and signaling pathways in the recipient cells have been investigated. EVs derived from MSCs at various stages of osteogenic differentiation contain altered microRNA profiles, with a specific set of osteogenesis-related microRNAs enriched in EVs from the late phases of osteogenic differentiation[139]. The upregulation of the pro-osteogenic microRNAs miR-10b and miR-21 and the downregulation of the anti-osteogenic microRNAs miR-31, miR-144, and miR-221 in EVs from late-differentiated MSCs paralleled the induction of osteogenic differentiation and mineralization by EVs[139–143]. This result indicated that MSC-derived EVs induced osteogenic differentiation via the transfer of osteogenesis-related microRNAs carried by the EVs. Additionally, EVs increased the expression of pro-osteogenic and pro-angiogenic miRNAs, miR-2861 and miR-210, in recipient MSCs, which corresponded to the increased expression of VEGF and the osteogenic master transcription factor RUNX2 and enhanced osteogenic differentiation[144]. In addition, the transfer of HIF-1 and miR-375 via EVs derived from MSCs enhanced the osteogenic differentiation of MSCs. Another study attributed the pro-osteogenic effects of MSC-derived EVs in part to the enrichment of Wnt3a in EVs and the targeting of the Wnt signaling pathway, one of the most well-known signaling pathways regulating osteogenic differentiation[145–147]. Zhang et al. demonstrated that MSC-derived EVs promoted osteogenic differentiation of MSCs by activating the PI3K/Akt signaling pathway, which has been shown to play crucial roles in osteoblast differentiation and bone formation[148–150].

**Exosome-producing cells in the articular microenvironment.** Articular chondrocytes and synoviocytes are the primary exosome-producing cells in joints. They participate in cell and tissue cross-talk by transporting a complex cargo of proteins, lipids, nucleic acids, and other molecules. Under normal conditions, exosomes preserve the equilibrium of the joint's microenvironment. Pathological conditions alter the composition and function of exosomes, which disrupts the equilibrium between anabolism and catabolism in articular chondrocytes and promotes inflammatory responses[151].

**Articular chondrocyte-derived exosomes.** The paucity of circulation and lymphatic system in cartilage tissue makes self-repair problematic. Instead, exosomes derived from typical chondrocytes assist in maintaining the stability of chondrocytes and their surrounding microenvironment. Chen et al. discovered that normal chondrocyte-derived exosomes promote cartilage formation by modulating chondrocyte precursor cells. Upon addition of chondrocyte exosomes to a tissue engineering scaffold, they stimulated the formation of regenerated cartilage via the TGF-/SMAD signaling pathway, where SOX-9 and COL-II were stably expressed[138]. Liu et al. found that exosomes derived from articular chondrocytes protected chondrocytes from destruction by downregulating inflammatory factors, promoted the synthesis of aggrecan and COL-II of ECM, and promoted the chondrogenic differentiation of C3H10T1/2 cells[152].

**Synoviocytes-derived exosomes.** Synoviocytes are commonly known as synovial fibroblasts (SFB) and synovial macrophages, with SFB constituting the predominant cell type in synovium tissues. Under normal conditions, exosomes derived from synoviocytes are released into the articular microenvironment and maintain homeostasis; however, under osteoarthritic conditions, exosomes derived from synoviocytes cause an imbalance in the anabolism and catabolism of chondrocytes in articular cartilage[153].

**A role for MSC-derived exosomes in immunomodulation.** MSC-derived EVs have been shown in preclinical models to inhibit TNFα-induced collagenase activity and improve cartilage regeneration in OA chondrocytes in vitro[154]. MSC-derived EVs have also been demonstrated to enhance the production of IL-10 by immature DCs, a key cytokine for suppressing inflammatory T-cell responses. Exosomes have been demonstrated in a CIA animal model to lower arthritis index, leukocyte infiltration, and, most critically, joint destruction[155]. These exosomes lowered the frequency of Th1 and Th17 cells by targeting STAT3 and T-bet with miRNA, hinting that they may be employed to treat arthritis[155]. Other researchers also discovered that using MSC-derived exosomes reduced the severity of CIA by reducing the pathogenic immune response. Mice who received this treatment had lower levels of IL-6 and TNF in their joints, higher levels of IL-10 in their spleen and lymph nodes, and a lower Th1/Th17 ratio[156]. According to earlier studies in CIA, exosomes may diminish inflammation by polarizing B lymphocytes into Breg-like cells[157]. Thus, evidence suggests that MSC-derived EVs can heal joint damage, mainly when delivered intra-articularly[158]. Multiple clinical trials on osteoarthritis and spinal cord injury using MSC-derived EVs in which several clusters of miRNA and their downstream cascades have been shown to perform important functions have been conducted[159]. According to these preclinical studies, MSC-derived EVs appear safe and scalable for clinical use. EVs' activity can also be boosted by changing their cargo or administering immunosuppressive cytokines like IL-10, which could boost anti-inflammatory and chondroprotective properties.

**A role of EVs in AxSpA.** Exosomes derived from T cells of AxSpA patients have the potential to modify the cytokine and expression profiles of normal T cells toward an inflammatory state by upregulation of RORγt, STAT3, T-bet, IL-17, IL-23, TNFα, and IFNγ[160]. In addition, destabilization of the articular microenvironment is accompanied by elevated levels of TNFα, IL-1, IL-6, and MMP-13, among others[161,162]. The tissues in the joint conform to the microenvironment of the joint. The articular microenvironment is therefore dysregulated or even disordered in AxSpA. In vivo cells secrete EVs into the microenvironment in order to influence it[163,164]. As stated previously, pathological osteogenesis and inflammation play a crucial role in AxSpA. Recent research sought to elucidate the mechanisms underlying the osteogenic differentiation of MSCs in AxSpA by EV-encapsulated miR-22-3p from M2 macrophages. Researchers demonstrate that the transfer of miR-22-3p by M2 macrophage-derived EVs promotes the progression of AxSpA via the PER2-mediated Wnt/-catenin axis; furthermore, the authors demonstrated that EVs-encapsulated miR-22-3p from M2 macrophages promote the osteogenic differentiation of MSCs[165]. Another research revealed that Circ-0110634 was expressed at a greater level in AxSpA patients' MSC-derived exosomes than in healthy exosomes. Circ-0110634 is a circular RNA that inhibits osteoclastogenesis when overexpressed[166].

Clinical studies are being conducted using three key sources of EVs: DCs, MSCs, and patient-derived tumor cells. Specifics of the cell culture process, the purification process, and EV quality control are all critical aspects of successful exosome manufacture. It is possible to direct miRNAs into exosomes utilizing the EXO-motif. miRNAs are, therefore, amenable to therapeutic modification of cellular functions. miRNAs in EVs are also protected from RNAse degradation, and their integrins and opsonins permit selective delivery of their intracellular content. As a proposed therapeutic approach, it has been hypothesized that the transfer of EVs derived from engineered MSC-derived EVs enriched with specific miRNA may have therapeutic potential of AxSpA[167]. EVs generated from genetically modified MSCs that are tailored to contain particular miRNAs might be used as molecular "Trojan horses" to selectively target recipient cells and improve immunotherapeutic responses. Furthermore, because MSC-derived EVs lack stimulatory HLA-complex molecules and surface co-stimulators, they do not cause adverse immunological responses, unlike native MSCs[168].

**Concluding remarks and future directions.** Long-standing conceptual and technological barriers have impeded MSC-derived EV research. When assigning particular functions to EVs, sensitivity and consistency are necessary despite the enormous advances in techniques for the separation and characterization of EVs. In addition, knowing the varied fates of EVs in recipient cells provides information on critical factors governing the distribution of functional cargo and will ultimately allow more efficient therapeutic use of EVs. Despite the fact that MSC-derived EVs elicit a lesser immune response and have a more acceptable safety profile than MSC cell therapy, their practical implementation still faces obstacles. EVs produced from MSCs are a potential cell-free therapy that may provide the therapeutic advantages of MSCs with fewer risks. The immunological responses of EVs generated from MSCs are mostly influenced by their miRNA and protein content. EVs have several biological applications because MSCs may be genetically modified to produce EVs containing targeted or therapeutic substances and

because these EVs can be chemically altered and loaded with cargo. To bring EVs products into clinical practice, it will also be necessary to overcome the present obstacles associated with the large-scale production of EVs in compliance with large-scale good manufacturing practice standards.

**Reporting summary**. Further information on research design is available in the Nature Portfolio Reporting Summary linked to this article.

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

## Acknowledgements
All figures (1–3) and every element of these images were created by F.T. under the supervision of R.I. All figures were created by Biorender and have been cited on the figures (created on Biorender.com).

## Author contributions
This review paper was written by F.T. under the supervision of R.I. at the Krembil Research Institute. R.I. approved the review and had direct supervision of this paper. All authors participated in the review of the manuscript.

## Competing interests
The authors declare no competing interests.
