## [Peer Review File · Communications Biology]

Reviewers' comments:

Reviewer #1 (Remarks to the Author):

This manuscript attempts to review the use of MSC exosomes for AxSpA. There are many factual errors and wrong or inappropriate citations such as citing review and not the scientific papers to support specific experimental observation or crediting a finding to a "me too" rather than the first report of the finding. Many statements were made without references or logic. Please refer to the comments in the attached manuscript pdf for specific examples. The comments are by no means exhaustive. According to the authors, the main rationale underpinning the use of MSC exosomes is the miRNA load in MSC exosomes. However, the rationale is poorly supported without due consideration of the current state of the art in field (please refer to the comments against this section in the attached manuscript pdf). There are simply too many comments to be listed here.

Reviewer #2 (Remarks to the Author):

This is a very well-written and comprehensive review on Mesenchymal stem cell in AS; the part relative to extracellular vesicles is not useful in the economy of the manuscript and do not add anything to the overall quality of the review. I would suggest removing this part, focusing only on MSC. In addition, I have some concerns to be addressed by the Authors:

1. A short introductory paragraph on HSC niche function(also including MSC) in general is required for those readers that are not familiar with this argument
2. A description of what "MSC" niche is and its physiological function is required
3. Since the Review is focused on AxSpA, a more comprehensive introduction on what the disease, on the possible importance of MSC in promoting the transition to fat lesions and new bone formation and the pathogenetic aspect of the disease is required
4. 10.3389/fgene.2022.947120. eCollection 2022; 10.3390/ijms23126660; 10.1155/2022/9463314; 10.1002/eji.202048878.: these manuscripts should be quoted and discussed
5. The role of MSC in influencing innate lymphoid cells differentiation should be discussed (10.3389/fimmu.2021.797432)
6. Since the recent description of inflammasome activation in AxSpA patients (10.1002/art.41644), the role of MSC in modulating inflammasome activation should be discussed
7. The metabolism of human mesenchymal stem cells during proliferation and differentiation should be discussed in relationship with the neo-apoptotic phenotype of AxSpA
8. A paragraph discussing the therapeutic options for AxSpA and their limitation is required to introduce the necessity of new therapy, such as MSC-based therapy

Response To Reviewers Letter

Dear Reviewers:

Thank you for the constructive comments concerning our manuscript entitled “Mesenchymal stem cell extracellular vesicles: Biology and therapeutic potential in Axial Spondyloarthritis”.

These comments were valuable and very helpful for improving our manuscript to better demonstrate the important significance of our research. We have carefully reviewed all comments and completed point-by-point revisions. The responses to the comments in this letter and the revised portions of the manuscript are marked in colors. We tried to delete/rewrite all the unclear sentences or the ones that have a lot of contradictory and uncertainty about them in the literature. All the missing references are added to the text. We appreciate the work of the Reviewers and hope that the revisions will meet with approval. We will be glad to respond to any further comments that you may have.

Yours sincerely,

Dr. Robert D. Inman

Schroeder Arthritis Institute, Toronto Western Hospital, 399 Bathurst Street, Room1E-423, Toronto, Ontario M5T 2S8, Canada.

Robert.Inman@uhn.ca

Reviewer #1 (Remarks to the Author):

1-The juxtaposition of this statement with the following statement regarding MSC suggests that MSCs can overcome the barriers that T cell therapy is facing. This statement is highly premature. First, no OTC MSC drugs have been created, in fact, no MSC drug has been approved by the US FDA. As such, the cost of developing MSC drugs is still unknown and technical difficulties remain intractable as evidenced by the fact that FDA has not approved a single MSC drug.

Response: We apologize for our unclear statement regarding MSCs therapy. Our main goal in this statement depended on MSCs expressing just a small number of class I MHC molecules and no class II MHC or costimulatory molecules (CD40, CD80, and CD86), and MSCs have a lower immunogenic potential but it was not clear in our previous statement. We rewrote the text to make it clearer and highlight it in the text. Although we decided to add the "Mesenchymal stem cell point of view: cell biology to therapeutic advancement" Section first to make the biology and function of the MSCs clear.

2- This reference is not appropriate. The multilineage potencies of MSCs were reported in the late 1990s.

Response: We apologize for the missing reference. We add the proper reference for this claim." Multilineage potential of adult human mesenchymal stem cells by MARK F. PITTENGER, 1999, Science".

3- These markers are not distinct to MSCs. Many fibroblasts express similar makers.

Response: Thank you for this constructive comment. We mentioned this critical point about MSC characterization in detail in the text and reference to minimal criteria for the definition of MSCs (Cytotherapy (2006) Vol. 8, No. 4, 315-317) which in addition to the surface antigens, mentioned the differentiation potential and plastic adherence as a minimal criterion to define MSC.

4- Human MSCs produce indoleamine-2,3-dioxygenase (IDO), which is one of the key immunosuppressive factors. This is not the right argument.

Response: Thank you for this constructive comment.

Although we have some references (inserted in the text) to support that human MSCs produce indoleamine-2,3-dioxygenase (IDO), which is one of the immunosuppressive factors. IDO degrades L-tryptophan, resulting in the local buildup of kynurenine, which suppresses T cell activation, proliferation, and functional activity, as well as Th17 differentiation.

As the reviewers mentioned, it has been reported that the immunomodulatory activity of MSCs was not compromised by a lack of IDO production caused by either defective IFN γ receptor 1 or IDO inhibitors (PMID: 17522338). The role of IDO in mediating the immunosuppressive activity of MSC is not definitive and remains controversial.

Also, the researchers evaluated the effects of adipose-derived stem cells (ASCs) on allergic inflammation in IDO-knockout (KO) asthmatic mice or asthmatic mice treated with ASCs derived from IDO-KO mice (PMID: 27812173). They found that IDO did not play a pivotal role in the suppression of allergic airway inflammation through ASCs, suggesting that IDO is not the major regulator responsible for suppressing allergic airway inflammation. Therefore, we discussed both views to have a balanced review.

5- This sentence is incomprehensible. Mammalian species, such as monkeys, pigs, dogs, cows, and humans, activate IDO, which is the predominant enzyme mediating MSC's immunoregulatory actions in rodents and is created by inflammatory cytokine.

Response: Thank you for this valuable comment. According to the comment by reviewers, we found this sentence so confusing so we decided to delete this sentence.

6- is this in reference to rodent or human MSC? If rodent MSC, what is the relevance of iNOS-NO axis to the use of MSC to treat AxSpA in human? At the minimum, there should be a discussion of the relative importance of iNOS and IDO in human versus rodent MSC.

Response: Based on the previous comment, this sentence is deleted to make the text clearer.

7- Given the fact that immunomodulatory activity of human MSCs was not compromised by a lack of IDO production caused by either defective IFN γ receptor 1 or IDO inhibitors (PMID: 17522338), this section is quite redundant or at least rationalize why this section is merited in this review.

Response: Thank you for this constructive comment. And based on the review's suggestion this section is deleted from the paper.

8- In the section "Therapy Using MSCs in AxSpA", the MSCs were not stimulated by cytokines prior to administration to patients. Clearly, the cytokine stimulation was not necessary for the immunomodulating activity of MSCs in the treatment of AxSpA. This is further evidenced by the

overwhelming majority of clinical trials that uses unstimulated MSCs. A more balanced review would be appreciated.

Response: Thank you for this helpful comment. We add the section of "Limitation of cytokine-stimulation of MSCs" and we tried to address the limitations of this method. For instance, it has been demonstrated that MSCs do not all react to stimulation in a coordinated manner to confer effective immunosuppression. Instead, a significant proportion of them have not yet established immunosuppressive activity or are in different phases of doing so after stimulation.

9- Contrary to the authors' claim, MSC products have received market approval in a few jurisdictions i.e. they have met safety and efficacy benchmarks:

May 17, 2012, Health Canada issued marketing approval for Prochymal™ to treat children with acute Graft versus Host Disease. September 15 2015 that the Japanese Ministry, Labour and Welfare has approved TEMCELL® for acute GvHD. March 2018, the European Commission has approved the first MSC pharmaceutical (Alofisel®) to treat Crohn's related enterocutaneous fistular disease.

Response: Thank you for this helpful comment. We tried to address this issue in the beginning section of "Therapy Using MSCs in AxSpA" in detail.

In the beginning part, we address that we have some MSC products with regulatory approval and we mentioned that several published clinical studies have shown the usefulness of MSCs as a therapeutic agent with little toxicity, and certain stem cell-based medications are licensed. In the final section, we address a crucial obstacle in using MSCs considering that as of January 2018, no MSC-based therapies have been authorized for diverse diseases, and there is not a single FDA-approved medication for use in the United States.

10-Are the authors implying that MSC differentiation is critical to their immune regulatory effects such that the prevention of their differentiation caused the reduction of their immune regulatory effects? This section should be discussed in the context of our current understanding of MSC homing, engrafting, and differentiation. It is now widely accepted that systemically administered MSC rarely home to injured tissues, engraft or differentiate. Even the authors acknowledged this in line 324. This entire section needs some serious consideration of the content.

Response: Thank you for this constructive comment. There was a study in RA which mentioned that the existence of blocking of both regenerative (differentiation) and immunomodulatory phenotypes under inflammatory conditions characterized by an upregulation of genes involved in immune processes and a simultaneous downregulation of genes mainly involved in regenerative or cell differentiation functions. And they conclude that the two main functions of MSCs (immunomodulation and differentiation) are blocked, at least while the inflammation is being resolved. Inflammation, at least partially mediated by gamma-interferon, drives MSCs to cellular distress adopting a defensive state (PMCID: PMC6501285). Although in the manuscript, this entire section has changed.

11-what studies? please provide references.

Response: We apologize for missing the reference. The proper reference is PMID: 22395436. Although, the entire section has changed in the manuscript.

12- This entire section needs some serious consideration of the content. Do MSCs create more problems than they resolve in arthritis?

Response: Thank you for this constructive comment. This section has completely changed.

13- The logic of this sentence is bewildering. If they are similar, why are adipose tissue MSCs more susceptible than BM-MSCs?

Response: We apologize for our unclear statement. This sentence is deleted.

14-This is scientifically baseless. many bioactive factors are secreted as soluble factors without being packaged into vesicles.

Response: We apologize for our unclear statement. This sentence is deleted.

15-This sentence imply that EVs is the only mode of intercellular communication for MSC when EV is only one of the ways for intercellular communication.

Response: We apologize for our unclear statement. We make it clear that "exosomes are only one of the ways for intercellular communication".

We explain in the text that exosomes operate as a mode of communication between and among cells and tissues is a powerful idea that if true, will change our understanding not only of human physiology but the practice of medicine.

We add "Bone remodeling in AxSpA" to provide a brief explanation of the physiology of osteoblasts (OB) and osteoclasts (OC) in AxSpA. Next, we mentioned the necessity of MSCs for maintaining bone homeostasis as they may undergo trilineage differentiation into OBs, chondroblasts, and adipoblasts to take part in bone remodeling and then we address the role of MSCs in AxSpA.

16-MVs have a larger size range which overlaps with that of exosomes, making it difficult to isolate them separately. Please refer to MISEV 2018 for more details and guidance on this subject.10.1080/20013078.2018.1535750

Response: We apologize for our unclear statement. We decided to keep the focus on exosomes rather than microvesicles because the main focus of this manuscript was talking about MSCs and MSC-exosomes. This sentence is deleted from the text.

17- where is the reference? There are many reports of monocytes or M0 macrophages polarizing to M2 after being exposed to MSC exosomes but not after transfection. The first report on M2 polarization by MSC exosome through TLR4 signaling was in 2014. Please provide a balanced literature review. what is the relevance of this statement to the previous statement and in the context of this section on extracellular vesicles?

Response: We apologize for our unclear statement. Because the reviewers mentioned that this section is un relevant to the Extracellular vesicles section, we removed it from this section. The entire section has changed.

18-The review reference cited definitely did not provide evidence that this is true. This sentence is definitely not true.

Response: Thank you for this constructive comment. The sentence is deleted.

19- what is the relevance of this statement to the previous statement and in the context of this section on extracellular vesicles?

Response: Thank you for this constructive comment. Based on the reviewer's previous comments we decide to delete this sentence.

20- please provide a reference. To date, there is no clinical trial on the use of MSC exosomes to treat GVHD. Only one clinical study was reported for a single patient who was treated on compassionate grounds.

Response: The sentence is deleted.

21- why are diabetic wounds being singled out? MSC exosome polarization of macrophages towards M2 have been reported earlier than the references cited here or in disease models that are more relevant to AxSpA. Why aren't these reports being cited and reviewed?

Response: Thank you for this constructive comment. This section was irrelevant and deleted.

22-How important or representative is this single report compared to the hundreds or even thousands of reports on the use of unmatched allogeneic MSC EVs and the use of human EVs in a wide range of immune-competent animals without immune suppression that resulted in positive therapeutic outcomes and without autoimmune responses?

Response: Thank you for this constructive comment. The entire section has changed in the manuscript.

23- Need more balanced perspective. There are many papers on MSC exosomes without pre-conditioning that have reported repairing of bone fractures

Response: We apologize for the unclear statement. The entire section has changed.

24- please cite a primary research paper and not a review paper.

Response: We apologize for missing the reference. The proper reference was added to the text.

25-There is NO clinical trials registered at clinicaltrial.gov in RA or OA for MSC exosomes.

Response: Thank you for this constructive comment. We make this statement clear in the text.

26- This statement regarding miRNA accounting for more than half of the exosome content is definitely not true. Furthermore, deep sequencing by two independent groups have revealed that miRNAs constitute 1-5% of the RNA cargo in MSC-sEVs. (<https://doi.org/10.3402/jev.v5.29828>; <https://doi.org/10.1186/s13287-015-0116-z>). It has been estimated theoretically and from experimental data, each sEV contains <1 copy of miRNA even for the most abundant miRNA (<https://doi.org/10.1073/pnas.1408301111>; <http://www.biochemsoctrans.org/content/ppbiost/46/4/843.full.pdf>). It was also proposed that based on the miRNA configuration, their maturity and accessory proteins for miRNA present in the MSC-sEVs, miRNAs are not likely to drive many of MSC-sEV activity (<http://www.biochemsoctrans.org/content/ppbiost/46/4/843.full.pdf>). In addition, the recent reports on the poor efficiency of EV uptake by cells, the escape of EVs from the endosome/lysosome pathway, the uncoating of EVs to release their contents in the cytosol (DOI:10.1038/s41467-021-22126-y; 10.1016/j.celrep.2022.110651; 10.1021/acsnano.9b10033)

have made the role of miRNA in driving MSC-EV activity increasingly untenable. This and the authors' own observation in line 324 that "Few transfused cells reach target organs because most become caught in the lungs and reticuloendothelial system" effectively demolish their argument for the importance of miRNA in driving MSC exosome activity, the contradicting reports should be mentioned and discussed for an objective and balanced review.

Response: Thank you for this constructive comment. We make it clear in the text that miRNA is not accounting for more than half of the exosome content. The reason we mentioned exosomal miRNA was to use exosomes as targeted vehicles to transfer candidate immunomodulatory miRNA to inflamed synovial tissue *in vivo* as a exosome based targeted delivery therapy. The use of MSC-derived exosomes can ground immunotherapy to overcome the difficulties associated with the clinical use of MSC. Currently, our lab is working on this project. For tissue-specific targeting of exosome *in vivo*, we will use MSCs-exosome expressing RGD fused to the integrin $\alpha V\beta 3$ to target synovial vasculature, leading to local delivery of their miRNA content. However, recent observations on the low efficiency of exosomes absorption by cells, the escape of exosomes from the endosome/lysosome pathway, and the uncoating of exosomes to release their contents in the cytosol have rendered the function of miRNA in regulating the activity of MSC-exosomes more questionable.

27- Not true. There is about 11 early phase trials using MSC exosomes/EVs registered at Clinicaltrial.gov. At least three have been completed and one/two already published.

Response: Thank you for this constructive comment. The whole section is rewrite.

28- when was the search conducted?

Response: We apologize for missing the date. The researched conducted on April 2022 based on the data provided by clinicalTrials.gov

29- is this a realistic probability?

Response: Thank you for this helpful comment. This sentence is deleted from the table.

30- possible with exosomes as well as these are similar in size to viral particles.

Response: Thank you for this constructive comment. It has been added to the exosomes part.

31- relative to MSCs?

Response: Thank you for this constructive comment. It has been deleted from this section.

Reviewer #2 (Remarks to the Author):

This is a very well-written and comprehensive review on Mesenchymal stem cell in AS; the part relative to extracellular vesicles is not useful in the economy of the manuscript and do not add anything to the overall quality of the review. I would suggest removing this part, focusing only on MSC. In addition, I have some concerns to be addressed by the Authors:

1. A short introductory paragraph on HSC niche function (also including MSC) in general is required for those readers that are not familiar with this argument

Response: Thank you for this constructive comment. MSCs represent crucial attributes of the bone marrow (BM) niche, where they interact with hematopoietic stem progenitor cells (HSPCs) by providing physical support and secreting soluble chemicals that govern HSPC maintenance and destiny. In addition, we add the "Mesenchymal stem cell point of view: cell biology to therapeutic advancement" Section first to make the biology and function of the MSCs clear.

2. A description of what “MSC” niche is and its physiological function is required

Response: Thank you for this constructive comment. We add this section to the manuscript. The niche of the human bone marrow (BM) consists of nonhematopoietic cells that provide physical support to hematopoietic stem progenitor cells (HSPCs) and maintain their homeostasis. MSCs are essential components of the bone marrow niche, where they provide newly formed osteoblasts for bone tissue regeneration and tightly regulate the fate of HSPCs through direct interaction and the secretion of soluble factors, thus attempting to play a crucial role in the development and differentiation of the hematopoietic system.

3. Since the Review is focused on AxSpA, a more comprehensive introduction on what the disease, on the possible importance of MSC in promoting the transition to fat lesions and new bone formation and the pathogenetic aspect of the disease is required.

Response: Thank you for this constructive comment. We add a section " Bone remodeling in AxSpA " in the text and try to explain the activity of osteoblasts (OB) and osteoclasts (OC), under normal physiological conditions and AxSpA, and explain the role of MSCs in bone remodeling.

4. 10.3389/fgene.2022.947120. eCollection 2022; 10.3390/ijms23126660; 10.1155/2022/9463314; 10.1002/eji.202048878.: these manuscripts should be quoted and discussed.

Response: Thank you for these helpful suggestions. We tried to discuss all the recommended papers in the text. We mentioned the role of DKK1, IL-17, and NET in the " MSCs in AxSpA" section. We mentioned the result of the Systematic Review and Meta-Analysis of Randomized Controlled Trial in the " Therapy Using MSCs in AxSpA" section.

5. The role of MSC in influencing innate lymphoid cell differentiation should be discussed (10.3389/fimmu.2021.797432).

Response: Thank you for these helpful suggestions. We tried to explain briefly about ILCs in the text. Consideration should be given to the effect of MSC on the differentiation of innate lymphoid cells (ILCs), which are necessary to sustain tissue homeostasis and bridges between the innate and adaptive immune systems, and may aid in the development of ILC-based therapies for inflammatory disorders.

6. Since the recent description of inflammasome activation in AxSpA patients (10.1002/art.41644), the role of MSC in modulating inflammasome activation should be discussed.

Response: Thank you for this constructive comment. We mentioned the role of the inflammasome in AxSpA patient pathogenesis. And we tried to provide some evidence of the role of MSC in modulating inflammasome activation in an autoimmune disease like RA.

7. The metabolism of human mesenchymal stem cells during proliferation and differentiation should be discussed in relationship with the neo-apoptotic phenotype of AxSpA.

Response: Thank you for this constructive comment.

WE add a section entitled " Mesenchymal stem cell point of view: cell biology to therapeutic advancement" and briefly explain about MSC characterization and features.

In addition, we add a section to this manuscript " Do MSCs create more problems than they resolve in AxSpA? " with two subtitles" Bone remodeling in AxSpA " and " MSCs in AxSpA " explaining the bone remodeling properties of MSCs in both normal and Inflammatory condition (like AxSpA)

8. A paragraph discussing the therapeutic options for AxSpA and their limitation is required to introduce the necessity of new therapy, such as MSC-based therapy.

Response: Thank you for these helpful suggestions. As the reviewer suggested we add " Limitations of MSC-based therapy for AxSpA" to the text to introduce the necessity of new therapy.

Reviewers' comments:

Reviewer #1 (Remarks to the Author):

Please refer to uploaded pdf

Reviewer #2 (Remarks to the Author):

Authors responded satisfactorily to Reviewer's comments. I recommend the publication of the review

Mesenchymal stem cells and their extracellular vesicles: Biology and therapeutic potential in Axial Spondyloarthritis

Fataneh Tavasolian¹, Robert D. Inman^{2,3,4*}

1-Spondylitis Program, Division of Rheumatology, Schroeder Arthritis Institute, University Health Network, Toronto, Ontario, Canada

2- Spondylitis Program, Division of Rheumatology, Schroeder Arthritis Institute, University Health Network, Toronto, Ontario, Canada

3-Krembil Research Institute, University Health Network, Toronto, Ontario, Canada

4-Departments of Medicine and Immunology, University of Toronto, Toronto, Ontario, Canada

*Corresponding author: Dr. Robert D. Inman, Schroeder Arthritis Institute, Toronto Western Hospital, 399 Bathurst Street, Room1E-423, Toronto, Ontario M5T 2S8, Canada.

Robert.Inman@uhn.ca

Abstract

Axial spondyloarthritis (AxSpA) is a chronic, inflammatory, autoimmune disease that predominantly affects the joints of the spine, causes chronic pain, and, in advanced stages, may result in spinal fusion. Recent developments in understanding the immunomodulatory and tissue-differentiating properties of mesenchymal stem cell (MSC) therapy have raised the possibility of applying such treatment to AxSpA. The therapeutic effectiveness of MSCs has been shown in numerous studies spanning a range of diseases. Several studies have been conducted examining acellular therapy based on MSC-secreted factors. **Compounds contained within extracellular vesicles (EVs) generated by MSCs have been proven to reproduce the impact of MSCs on target cells. These EVs are associated with immunological regulation, tissue remodeling, and cellular homeostasis.** EVs biological effects rely on their cargo, with microRNAs (miRNAs) integrated into EVs playing a particularly important role in gene expression regulation. In this article, we will discuss the impact of EVs generated by MSCs on target cells and how this may be used as a unique treatment strategy for AxSpA.

Keywords: Mesenchymal stem cell, extracellular vesicles, exosomes, Axial Spondyloarthritis

Introduction

AxSpA is an autoimmune disease resulting from a complicated interplay between genetics and the environment[1]. Despite breakthroughs over the last several decades, the etiology of AxSpA remains unknown. Genetic background, immunological response, microbial infection, and endocrine dysregulation are some of the factors linked to the development of AxSpA in previous studies[2]. HLA-B27 has been associated with AxSpA in numerous

populations worldwide, but the mechanisms remain unclear[3]. HLA-B27 is present in 90–95% of AxSpA patients, although only 1%–2% of HLA-B27-positive individuals will develop AxSpA [4]. HLA-B27 is found on the cell surface in both heterodimeric and homodimeric forms and intracellular and exosomal MHC-I dimers[5]. In recent years, HLA non-B27 and non-HLA genes have been identified in AxSpA as a result of genome-wide association studies (GWAS) and other technologies. ERAP1 and IL23R provided early evidence for an important role for numerous non-MHC genes associated with AxSpA [6, 7].

Factors affecting the immune system and microbes

AxSpA has been linked to a variety of autoimmune diseases, such as IBD, anterior uveitis, and psoriasis, implicating a common genetic and immunological basis. **Recent study indicates that mononuclear cells of AxSpA patients generate IL-23 and IL-17 in response to inflammasome activation[8]. Inflammasomes are receptors and sensors of the innate immune system that organise antimicrobial host defenses and control the inflammatory response[9].** Also the researchers demonstrate that synovial fluid (SF) in AxSpA patients is enriched with CD8+ CTLs with distinctive integrin expression patterns that suggest gut-joint trafficking[10, 11]. Microbial infection may act as a catalyst for the development of AxSpA by activating the host's innate immune system. HLA-B27 transgenic rats do not develop AxSpA symptoms in a germ-free environment, but this changed when commensal bacteria were administered to the germ-free animals, revealing possible links between HLA-B27 and the microbiome. The gut microbiome is critical for gut homeostasis, and its disruption has major implications for immune control and the progression of AxSpA disease[7, 12]. Although it is now accepted that individuals with AxSpA have an altered gut microbiome, no consistent and uniform pathogen profile has been identified. The gut microbiome is thought to have a role in AxSpA through interacting with genetic, immune-mediated, and microbial metabolic disorders [11].

Inflammation and new bone formation in AxSpA

Axial and peripheral joints often exhibit inflammation and the formation of new bone. The most often affected areas are the spinal vertebral corners, sacroiliac joints, sternal rib junction, iliac crest, ischial tuberosity, Achilles tendon, and plantar fascia[13, 14]. Despite the fact that magnetic resonance imaging scans may identify inflammation before the development of radiographic abnormalities and that early pharmaceutical intervention can reduce inflammation, successfully preventing the formation of new bone remains a tough challenge. The finding of components responsible for new bone formation might pave the way for the creation of novel therapeutics for AS. It is believed that genetics, immune cell interactions, inflammatory cytokines, and anabolic signaling pathways influence inflammation and subsequent bone formation[15].

Several studies have shown the possibility of osteogenic differentiation of MSCs from AxSpA patients using recent technical advances[3, 16-19]. These findings, together with the therapeutic efficacy of MSCs infusion in both preclinical and clinical studies, suggest that MSCs may play a dual role in the development and treatment of AxSpA. This review summarises the MSCs cell biology and their therapeutic advancement, analyses the many roles of MSCs in the development and treatment of AxSpA, enumerates the current clinical MSC trials in AxSpA diseases, and presents EVs derived MSCs as a possible innovative therapy for AxSpA.

Mesenchymal stem cell point of view: cell biology to therapeutic advancement

While much research on cell-based therapies has focused on regenerative medicine, there remains the hope of employing cell therapies as new, alternative treatments for various diseases. Human multipotent MSCs are now being studied for their possible application in therapy for several incurable diseases. Due to their capacity for differentiation, immunomodulation, and paracrine factor release, MSCs have attracted considerable attention as candidates for cellular therapies[20]. MSCs are non-hematopoietic cell progenitors first isolated in bone marrow but now recognized to be present in a variety of other tissues. They possess the capacity for self-renewal and display limited differentiation. [21-23] (Figure 1). MSCs may be found in the stroma of all adult organs, although they are most often found in perivascular areas, where they contribute to tissue homeostasis, monitoring, repair, and remodeling. **The niche of the human bone marrow (BM) consists of nonhematopoietic cells that provide physical support to hematopoietic stem progenitor cells (HSPCs) and maintain their homeostasis. MSCs are essential components of the bone marrow niche, where they provide newly formed osteoblasts for bone tissue regeneration and tightly regulate the fate of HSPCs through direct interaction and the secretion of soluble factors, thus attempting to play a crucial role in the development and differentiation of the hematopoietic system.**[24-26]. Researchers have examined MSCs utilizing diverse separation and development procedures to characterize the cells. Comparing and contrasting studies becomes more challenging. In order to begin addressing this issue, the Mesenchymal and Tissue Stem Cell Committee of the International Society for Cellular Therapy has established basic criteria for characterizing human MSCs. First, the MSCs must adhere to plastic when grown under standard circumstances. MSCs must also express the surface molecules CD105, CD73, and CD90, but not CD45, CD34, CD14, CD11b, CD79, CD19, or HLA-DR. Third, MSCs must be capable of differentiating into osteoblasts, adipocytes, and chondroblasts in vitro. Despite the likelihood that these criteria may need to be modified in the future, scientists anticipate that this basic set of standard criteria will result in a more consistent categorization of MSCs and will simplify the sharing of data across researchers[27, 28]. Because of their anti-inflammatory, immunomodulatory, and regenerative capabilities, MSCs have been studied for use in cell-based therapeutics[29]. Paracrine and cell-to-cell contact pathways mediate these responses. Paracrine effects are mediated by the MSC secretome, which comprises cytokines, chemokines, and microvesicles/exosomes that transport proteins or miRNAs to target cells[27]. The MSC secretome is rich in immunoregulatory factors that may affect innate and adaptive immune responses. TGFB-1, hepatocyte growth factor (HGF), prostaglandin-E2 (PGE2), interleukin-6 (IL-6), interleukin-10 (IL-10), nitric oxide (NO), human leukocyte antigen-G5 molecules (HLA-G5), and leukemia inhibitory factor (LIF) are among the molecules generated by MSCs (LIF)[30]. The specific processes by which these factors suppress or alter immune cells are incompletely understood. The development and expansion of immune regulatory cells is another mechanism by which MSCs can modulate the immune response. Under homeostatic conditions, MSCs express just a small number of class I MHC molecules and no class II MHC or costimulatory molecules (CD40, CD80, and CD86)[31, 32]. MSCs are hypoimmunogenic under homeostatic settings as a consequence, making them appropriate for allogeneic transplantation. Chemokine receptors, matrix metalloproteinases (MMPs), and adhesion molecules are all abundantly expressed in MSCs, which might contribute to their migration to sites of inflammation. [33, 34].

Do MSCs create more problems than they resolve in AxSpA?

Bone remodeling in AxSpA

The coupling activity of osteoblasts (OB) and osteoclasts (OC), under normal physiological conditions, maintain the dynamic equilibrium of bone formation and bone resorption[35-37]. In contrast to OC, which orchestrate bone loss, OB generate an organic matrix and aids mineralization. The interactions between cells tightly regulate this bone remodeling process. For bones to maintain their mechanical integrity and strength, synthesis and absorption must be appropriately balanced [38]. OC and OB activities, however, become uncoupled under inflammatory situations, leading to excessive bone resorption or formation[39]. The degradation of bone by osteolysis and osteogenesis coexist in AxSpA. Early inflammatory lesions of localized hyperemia and edema are associated with OB activity and bone marrow edema is often significant when radiographic signs of joint injury have not yet appeared[40, 41]. As a result, more OCs saw on AxSpA radiographs during acute inflammation.

Over time, chronic inflammation induces an anabolic skeletal reaction, with new cortical bone synthesis at sites of inflammation. This can be associated with excessive trabecular bone resorption; and the trabecular bone loss is frequently found to be connected to the formation of new bone at the enthesis sites[41, 42]. MSCs have a higher capacity for osteogenic differentiation at this time, and OBs form more ossification foci in the subchondral granulation tissue, which may precede the formation of marginal syndesmophytes. Therefore, the degree of local bone inflammation in the spine is thought to be the precursor of spinal radiographic damage in AxSpA [3, 43].

The complex interaction of cytokines and signaling pathways released by various immune cells on bone cell activity and bone mass has been increasingly clarified. The immune system modulates distinct bone cell types differently at different stages of the disease, including the acute, chronic, progressive, and therapeutic phases[36, 43, 44].

MSCs demonstrate enhanced osteoblast differentiation in AxSpA.

MSCs are essential for maintaining bone homeostasis as they may undergo trilineage differentiation into OBs, chondroblasts, and adipoblasts to take part in bone remodeling[45]. Due to their immunomodulation abilities, including their capacity for self-renewal and multipotent differentiation, MSCs may play a role in AxSpA [46, 47]. **MSC osteogenic development is controlled by several intracellular signalling networks, including the BMP/Smad pathway, the WNT/catenin pathway, and the MAPK system. These signaling pathways also play a role in the pathological osteogenesis associated with AxSpA. Numerous research investigate the reason for the increased osteogenic differentiation potential of MSCs in AxSpA[48].** BM-MSCs play a vital role in healthy joints by preserving bone homeostasis and repairing damaged tissues. However, selective RANKL expression in MSCs may contribute to joint inflammation in an inflammatory environment. In AxSpA this could result in binding of RANKL to RANK in inflammatory MSCs, thereby contributing to reverse signaling in osteoblasts and promotion of osteoblast differentiation (Figure 2) [49-52].

BM-MSCs from AxSpA patients have a greater intrinsic ability for osteogenic growth than BM-MSCs from healthy donors[18]. An imbalance between enhanced BMP-2 and decreased Noggin secretion was connected to AxSpA-MSC osteogenic differentiation, according to studies comparing the osteogenic differentiation capability of sternal BM-MSCs from AxSpA to healthy donors[3]. In AxSpA patients, BMP2 expression was considerably greater in BM-MSCs from ossifying entheses. Increased osteogenic differentiation is a consequence of BMP2 overexpression[3]. MCP1 is another factor which **MSCs generated more during abnormal osteogenic differentiation in AxSpA** and induces monocyte dysfunctions. Therefore, aberrant osteogenesis might result in AxSpA inflammation[19]. **Additionally, it is recognized that the immunomodulatory capacity of MSCs from AxSpA patients is decreased, possibly due to an imbalance between CCR4 + CCR6 + Th/Treg cells[53]. Different cytokine concentrations affect MSC regulation at various levels [54]. For instance,**

IL-17 is elevated in AS patients and suppresses DKK-1 expression while promoting osteoblastic activity, as observed by the researchers[55]. Dickkopf-1 (Dkk-1) is an essential regulator of bone remodelling in spondyloarthropathies. The expression of IL-17A on neutrophil extracellular traps stimulates the osteogenic potential of MSCs[17].

while low levels of IL-17A promote polarization of TLR4+ MSC and inhibit osteogenic differentiation via the JAK2/STAT3 pathway, high levels of IL-17A promote TLR3+ MSC polarization and enhance osteogenic differentiation via the Wnt10b/Runx2 pathway[56]. Control of MSC apoptosis is also a key factor and MSCs from AxSpA patients exhibit greater levels of apoptosis than healthy MSCs[57]. This is likely because MSCs elicit effector T cells by secreting chemokines that either mediate direct immunoregulation or cause Fas /FASL-induced apoptosis[57, 58]. TNF-related apoptosis-inducing ligand receptor 2 (TRAIL-R2) is expressed at higher levels in MSCs from AxSpA patients than in healthy MSCs, rendering them more vulnerable to TNF/CHX-induced apoptosis[44, 59]. MSCs from AxSpA patients were shown to elicit TNF-mediated inflammatory responses and higher osteogenic differentiation[60]. In active AxSpA, the frequency of Treg and Foxp3+ cells was reduced, whereas the frequency of CCR4+CCR6+ Th cells rose, indicating that those BM-MSCs had a poorer immunomodulatory potential[53]. MSCs from patients with AxSpA show lower immunoregulatory function[19, 53, 60, 61].

Therapy Using external MSCs in AxSpA

External MSC implantations have been proven to have positive and protective effects on AxSpA diseases in both preclinical and clinical investigations. These MSCs are amplified in vitro from either autologous (originating from the same person) or allogeneic (originating from the same species but not the same individual) sources. MSCs may be directly injected into an inflammatory joint [62, 63], and if this is not feasible, cells may be delivered into the body by a systemic injection, in which case external MSCs with homing ability may migrate to inflamed sites. How MSCs from internal and exterior sources act differently throughout the formation of AxSpA is fascinating. The key to answering this question may lie in the milieu that the internal MSCs encountered when entrapped in damaged tissue and that the external MSCs directly encountered during circulation or in synovial tissues following cell injection.

In the presence or absence of IL-23, enthesitis in AxSpA has a high number of immune cells that produce type 3 immunity-related cytokines (IL-17, IL-22, and GM-CSF)[64-66]. In

concert with other potential risk factors, such as male gender, HLA-B27 status, and mechanical loading stress, these cytokines initiate and sustain inflammation in spinal entheses, resulting in the induction of new bone formation[67, 68]. MIF and TNF are likewise released largely by myeloid cells and cause osteoproliferative alterations[69]. In response to new bone formation-initiating stimuli, osteo-chondro-progenitor cells, such as mesenchymal stem cells or periosteal cells, differentiate into osteoblasts or chondrocytes to produce new bone by intramembranous or endochondral ossification, respectively. During differentiation, crucial anabolic molecules and signalling pathways are active, including BMPs, RANKL, and Wnt. These findings demonstrate that MSCs are exposed to a diverse microenvironment, and the diversity of environmental stimuli may result in a wide range of cellular responses[15, 69].

Clinical trials of MSCs in AxSpA

Many clinical trials examining MSC transplantation in rheumatic diseases are now underway, including phase I/II studies in AxSpA to determine the safety and therapeutic advantages of MSCs transplantation[70]. MSCs transplantation has been offered as a treatment option for AxSpA patients who cannot use anti-inflammatory medicines because of their immune modulation abilities. The number of Treg cells in AxSpA patients is low in prior studies. MSCs may also prevent Th17 cell production by prompting T cells to develop into the Treg phenotype, decreasing the number of Th17 cells[71-73]. In a phase 1 clinical trial, human umbilical cord-derived MSCs (hUC-MSCs) were administered IV and repeated after three months, in combination with DMARDs (NCT01420432). In another trial, patients with AxSpA received infusions of human MSCs and NSAIDs (NCT01709656). A phase 2 clinical study (NCT02809781) is now underway to assess the use of human bone marrow-derived MSCs in AxSpA patients, as well as a phase I/II clinical trial (ChiCTR-TRC-11001417) to determine the safety of MSC treatment in AxSpA patients [62]. However, there is less information on their effectiveness. A 20-week clinical trial using allogenic MSCs administered IV was done with AxSpA patients who had failed to respond to NSAIDs. The absence of a control group necessitates further data, despite the fact that this research seems to be a promising therapy for patients[74]. **A result derived from a meta-analysis of randomized controlled trials Research reveals that six months of MSC treatment for AxSpA may increase the overall effective rate, reduce erythrocyte sedimentation rate, intercellular adhesion molecules, and serum TNF[75].** In addition Consideration should be given to the effect of MSC on the differentiation of innate lymphoid cells (ILCs), which are necessary to sustain tissue homeostasis and bridges between the innate and adaptive immune systems, and may aid in the development of ILC-based therapies for inflammatory disorders[76, 77].

Limitations of MSC-based therapy for AxSpA

Because it is challenging to collect MSCs from enthesal BM, most investigations on BM-MSCs from AxSpA patients have employed BM-MSCs from distant regions (such as the sternum) or produced pluripotent stem cells (such as dermal fibroblasts)[14, 19, 78]. Numerous cell-delivery strategies are ineffective, with many studies demonstrating that only a tiny fraction of injected cells stay at the injection site days after transplantation[79]. Few transfused cells reach target organs because most become caught in the lungs and reticuloendothelial system [71]. Although the clinical trial design is receiving attention, the specific equipment and techniques used to implant the cells locally have lately been profiled

more[80]. The remaining challenges include prices and potential adverse effects, which might lead to preclinical and clinical testing inconsistencies. Differential cell behavior, dose and distribution accuracy, and cell retention and survival after injection are only a few hurdles that must be overcome before meaningful translation can occur. The success of injectable cell transplantation depends on accurate measurement of post-injection cellular health and the development of consistent delivery mechanisms[81]. Consequently, prospective controlled trials are considered necessary to measure MSC-based therapy and determine its potential efficacy, specifically in treating AxSpA [82].

Nonetheless, as of January 2018, no MSC-based therapies have been authorized for diverse diseases, and there is not a single FDA-approved medication for use in the United States[83-86]. A crucial obstacle is guaranteeing that the MSCs when supplied to patients, will execute the same targeted function in concert. MSCs are very sensitive to their surroundings. In a lab-based production method, MSCs are exposed to an environment different from the human body, which might alter their response to growth factors and produce MSC preparations introducing unexpected and unwanted variability. Additionally, the behavior of the cells may change after being implanted into a patient. For example, they may not effectively reduce inflammation, nor produce tissue in undesirable locations, nor form structurally sound tissue. Cell and nuclear morphology may serve as possible distinguishing characteristics of MSC potency[87, 88]. The influence of morphology-directed stem cell lineage determination has been shown in both 2D and 3D and may serve as an early signal of osteogenic differentiation for MSCs[88-91]. It has also been established that the size of MSCs increases with passage and donor age; hence it is feasible that underlying morphological distinctions in MSC populations might explain or predict their variability in potency[92-94]. Like cell morphology, nuclear morphology has been recognized as a predictor of stem cell activity and a phenotypic indicator of epigenetic and transcriptional cellular activities[95].

MSC-exosome: A novel cell-free therapy

Extracellular vesicles

Extracellular vesicles (EVs) are tiny vesicles generated by almost all cell types, characterised by a phospholipid bilayer and harbouring a wide array of proteins, mRNAs, and miRNAs. Exosomes, (diameter less than 150 nm), are formed in the endosomal compartment in so-called multivesicular bodies, and microvesicles, or microparticles (diameters range from 150 to 1000 nm), are released by plasma membrane budding.[96]. The International Society of Extracellular Vesicles has published basic criteria to characterise EVs, including shape, the process of cellular release, and biochemical characteristics [97-99]. Therapeutic effectiveness of MSCs-derived EVs (MSC-EVs) has been described in several disease models, including myocardial infarction and reperfusion damage, liver and kidney injury, hind limb ischemia, and inflammatory illnesses . Although there is considerable interest in MSC-EVs for the therapy of several illnesses, little is known about their precise function[100, 101].

Exosomes

Exosomes are the most well-studied EV subclass. Exosome membranes are distinguished from endosomes by the presence of lipid rafts, which are involved in the fusion and release of intraluminal vesicles (ILV) and multivesicular bodies (MVB). MVB attaches to the plasma membrane, and exosomes are released. Membrane fusion, endocytosis, or cell type-specific phagocytosis may then be used by other cells to pick up exosomes [102]. The ability of exosomes to carry microRNAs, lipids, and proteins through tissue and biological barriers makes them promising as therapeutic vehicles (Figure 3). [103]. The emerging consensus that exosomes operate as a mode of communication between and among cells and tissues is a powerful idea that if true, will change our understanding not only of human physiology but the practice of medicine[104].It has been demonstrated that the protein profiles of serum-derived exosomes differed between AxSpA patients and healthy subjects. In a functional analysis, the differentially expressed proteins may contribute to alteration in immune responses. Differentially expressed proteins have been discovered in AxSpA serum-derived exosomes, which may provide new insights into the pathophysiology of AxSpA and could lead to the discovery of novel biomarkers for the disease[105-108].

Microvesicles

When a cell is stimulated, triggered or undergoes apoptosis, microvesicles are released by the outward budding and fission of the plasma membrane, while exosomes are produced via the inward budding of the limiting membrane of early endosomes[109]. In spite of this, they share traits of excellent biocompatibility, minimal immunogenicity, and targeting, and may be employed as drug carriers. The use of microvesicles produced from tumour cells to transport chemotherapy medications has been found to improve cancer treatment outcomes with few unwanted effects[109, 110].

Therapeutics based on MSC-derived EVs

EVs are one of the ways for intercellular communication. For cell-free MSC-based therapies, EVs have come to prominence as a novel therapeutic paradigm[111]. [112-114]. MSC-derived EVs have also been shown to reduce lung damage and the advancement of renal fibrosis by altering the phenotype and function of invading macrophages[115, 116]. MSC-derived EVs are just as successful as MSCs in treating a wide range of degenerative diseases and immunological issues but without the drawbacks of direct cell infusion. This presents the attractive possibility that the therapeutic activity of human MSCs may be mimicked by using their respective EVs. But developing more effective MSC- derived EVs based therapies for disease treatment remains a major challenge in the clinical use of MSCs. MSC- derived EVs have a number of advantages as therapeutic carriers, including the fact that they are mostly non-immunogenic and may migrate to remote sites in situations where inflammation is widespread. MSC- derived EVs based treatments have shown no toxicity or adverse immune response in animals injected with either native or modified exosomes, supporting their safety in these settings[117]. However, in order to find the most effective technique for delivering favorable therapeutic outcomes, the specifics of administration and frequency of EVs injections into patients must be addressed.

MSC- derived EVs have been shown in preclinical models to inhibit TNF α -induced collagenase activity and improve cartilage regeneration in OA chondrocytes in vitro[118]. MSC- derived EVs have also been demonstrated to enhance the production of IL-10 by immature DCs, a key cytokine for suppressing inflammatory T-cell responses. Exosomes have been demonstrated in a CIA animal model to lower arthritis index, leukocyte

infiltration, and, most critically, joint destruction[119]. These exosomes lowered the frequency of Th1 and Th17 cells by targeting STAT3 and T-bet with miRNA, hinting that they may be employed to treat arthritis[119]. Other researchers also discovered that using MSCs-derived exosomes reduced the severity of CIA by reducing the pathogenic immune response. Mice who received this treatment had lower levels of IL-6 and TNF in their joints, higher levels of IL-10 in their spleen and lymph nodes, and a lower Th1/Th17 ratio[120]. According to earlier studies in CIA, exosomes may diminish inflammation by polarizing B lymphocytes into Breg-like cells[121]. Thus, evidence suggests that MSC-derived EVs can heal joint damage, mainly when delivered intra-articularly[122]. Multiple clinical trials on osteoarthritis and spinal cord injury using MSC-derived EVs in which several clusters of miRNA and their downstream cascades have been shown to perform important functions have been conducted[123]. According to these preclinical studies, MSC-derived EVs appear safe and scalable for clinical use. EVs activity can also be boosted by changing their cargo or administering immunosuppressive cytokines like IL-10, which could boost anti-inflammatory and chondroprotective properties.

Furthermore, genetic modification of MSCs has been shown to improve the immunosuppressive and chondroprotective properties of their EVs. The cargo of EVs, which can contain a variety of chemicals, has a direct impact on their therapeutic effect on the target cell. MiRNAs are small non-coding RNAs that regulate gene expression by binding to the 3'-UTR of targeted mRNAs and by blocking post-transcriptional translation of the gene with which they are associated. Post-transcriptional processes, unlike transcriptional and epigenetic control, may fine-tune cell fate choices in response to environmental signals much faster[124, 125]. MiRNAs in EVs are also shielded from RNase destruction, and their integrins and opsonins allow for selective delivery of their internal content [126-129]. The advantages and disadvantages of utilizing MSCs and MSC-derived EVs are summarized in Table 1. **Through the miR-4284/CXCL5 axis, it has been shown that MSCs decrease osteoclastogenesis in an aberrant way in AxSpA [130]. In addition other group demonstrate that the transfer of miR-22-3p by M2 macrophage derived EVs promotes the progression of AxSpA via the PER2-mediated Wnt/-catenin axis; furthermore, the authors demonstrated that EVs-encapsulated miR-22-3p from M2 macrophages promote the osteogenic differentiation of MSCs[131]; As a proposed therapeutic approach, we hypothesise that the transfer of EVs derived from engineered MSCs may prevent the development of AxSpA. [132]. According to a research, MiR-204-5p suppresses the osteogenic development of AxSpA fibroblasts by modulating the Notch2 signalling pathway[133], It has been shown that miR-495 inhibits inflammatory response and promotes bone development of fibroblast-like synovial cells in AxSpA, while also modulating the wnt/-catenin/Runx-2 pathway via targeting inflammatory factors and dishevelled 2[134]. According to the results of the researchers, the level of miR-17-5p was significantly higher in the fibroblasts and ligament tissues of AxSpA patients compared to non- AxSpA individuals. When miR-17-5p was knocked down in fibroblasts taken from AxSpA patients, osteogenic differentiation and ossification were diminished. In contrast, fibroblasts isolated from AxSpA patients with elevated miR-17-5p levels displayed increased osteogenesis. In addition, inhibiting miR-17-5p reduced osteophyte formation and emulsified collagen reduced the sacroiliitis phenotype in rats with AxSpA. MiR-17-5p modulated the mechanism of osteogenic differentiation by targeting the 3 UTR of ankylosis protein homolog (ANKH)[135]. and other researchers have shown that adipose tissue-derived MSCs that overexpress miR-21 may alleviate spine osteoporosis in AxSpA mice[136]. EVs generated from genetically modified MSCs that are tailored to contain particular miRNAs might be used as molecular "Trojan horses" to selectively target recipient cells and improve**

immunotherapeutic responses. Furthermore, because MSC- derived EVs lack stimulatory HLA-complex molecules and surface co-stimulators, they do not cause adverse immunological responses, unlike native MSCs (Figure 5)[137]. Clinical studies are being conducted using three key sources of EVs: DCs, MSCs, and patient-derived tumor cells. Specifics of the cell culture process, the purification process, and EVs quality control are all critical aspects of successful exosome manufacture.

Several miRNAs have been associated with AxSpA -related events, including inflammation (miR-16, miR-221, miR-196a2, let-7i, and miR-96) (103,108), new bone formation (miR-20a, mir-29a, miR-300, miR-185, miR-30d, miR-320a, miR-130b, miR-33a, miR-155, and miR-222), and T cell modulation (m (Let-7i, miR-16, miR-124)[12, 131, 138]. The therapeutic potential of miRNAs in AxSpA shows great promise, and their administration through MSC- derived EVs may significantly accelerate the transition to clinical trials. To assess the safety of miRNA treatment, further study is required to identify the full effect of miRNAs on target cells and other types of cells[104, 129]. *As a novel EVs-based targeted delivery treatment, there is a potential to use MSC-derived EVs as targeted carriers to transport candidate immunomodulatory miRNA to inflamed synovial tissue. However, recent observations on the low efficiency of EVs absorption by cells, the escape of EVs from the endosome/lysosome pathway, and the uncoating of EVs to release their contents in the cytosol have rendered the function of miRNA in regulating the activity of MSC- derived EVs more questionable.[139-141].*

Concluding Remarks and Future Directions

Long-standing conceptual and technological barriers have impeded MSC- derived EVs research. When assigning particular functions to EVs, sensitivity and consistency are necessary despite the enormous advances in techniques for the separation and characterization of EVs. In addition, knowing the varied fates of EVs in recipient cells provides information on critical factors governing the distribution of functional cargo and will ultimately allow more efficient therapeutic use of EVs. Despite the fact that MSC-derived EVs elicit a lesser immune response and have more acceptable safety profile than MSC cell therapy, their practical implementation still faces obstacles. EVs produced from MSCs are a potential cell-free therapy that may provide the therapeutic advantages of MSCs with fewer risks. The immunological responses of EVs generated from MSCs are mostly influenced by their miRNA and protein content. Although techniques exist for identifying miRNAs in EVs, the key target genes of miRNAs produced from EVs remain mostly unexplored. EVs have several biological applications because MSCs may be genetically modified to produce EVs containing targeted or therapeutic substances and because these EVs can be chemically altered and loaded with cargo. To bring EVs products into clinical practice, it will also be necessary to overcome the present obstacles associated with the large-scale production of EVs in compliance with Current Good Manufacturing Practice standards. There are still unanswered concerns about the relative importance of EVs-associated vs. soluble mediators and the role of EVs in epigenetic and metabolic alterations at the single-cell level of immunity.

Figure 1: The properties and applications of mesenchymal stem cells. MSCs are composed of multipotent stem/progenitor cells. Under certain conditions *in vitro* and *in vivo*, MSCs may differentiate into distinct lineages. MSCs exhibit significant anti-inflammatory and immune-modulating properties.

Figure 2: The RANKL-RANK signaling functioning model that is suggested controls osteoblast differentiation and bone formation in inflammatory conditions in AxSpA. BM-MSCs provide a crucial job in normal joints by maintaining bone homeostasis and repairing damaged lesions due to their distinct normal environment activities. However, selective RANKL expression in MSCs may contribute to joint inflammation in an inflammatory environment. MSCs are thus candidate target cells for TNF in these disorders. Different cells secreting IL-22 in entheses provide an additional option for the involvement of BM-MSCs in AxSpA. The majority of studies evaluating the function of BM-MSCs in the pathophysiology of AxSpA have focused on their engagement in the ossification of entheses, which is characteristic of persistent AxSpA. In inflammatory MSCs expressing RANK, RANKL binding to RANK stimulates RANKL to reverse signaling in osteoblasts and promotes osteoblast differentiation[49].

Figure 3: Exosome characterization, isolation from MSC, and application as novel gene therapy. Exosomes are cell-secreted nanoparticles (30–150 nm in size) containing various biological components, including nucleic acids, proteins, and lipids, which play crucial roles in intercellular communication. As carriers, exosomes offer promise as enhanced platforms for targeted gene delivery due to their unique features, including intrinsic stability, minimal immunogenicity, and exceptional tissue/cell penetration potential. Targeted delivery raises the local concentration of therapeutics while minimizing negative effects.

Table 1. MSCs and MSC-exosomes for therapeutic applications: benefits and drawbacks

	Positive aspects	Negative aspects
MSCs	simple to isolate and collect	Probability of transmitting infections
	Highly prolific	Concerns over the associated regenerative process
	multilineage differentiation	
	Limited likelihood of immunological issues	
	cumulated experimental and clinical outcomes	
MSC-exosomes	Effectiveness via particular proteins in the exosome membranes and natural homing capability	A minimal isolation procedure is indicated
	Low chance of teratoma development	No controlled production procedures
	Excellent medication delivery system for both hydrophobic and hydrophilic substances	Quick elimination from the bloodstream upon injection (in vivo)
	Unaffected by freezing and thawing (compared with cells)	Challenges in isolating and purifying exosomes containing certain bioactive compounds
	Paracrine function	Deficiency of methods and tools to precisely characterize the chemical and physical characteristics of exosomes
		Minimal and restricted investigations on exosome-based treatments
		Probability of transmitting infections

References:

1. Zhu W, He X, Cheng K et al. Ankylosing spondylitis: etiology, pathogenesis, and treatments. *Bone research* 2019; 7: 1-16.
2. Hwang MC, Ridley L, Reveille JD. Ankylosing spondylitis risk factors: a systematic literature review. *Clinical Rheumatology* 2021; 40: 3079-3093.
3. Zheng G, Xie Z, Wang P et al. Enhanced osteogenic differentiation of mesenchymal stem cells in ankylosing spondylitis: a study based on a three-dimensional biomimetic environment. *Cell death & disease* 2019; 10: 1-11.
4. Rosenbaum JT, Weisman MH, Hamilton H et al. HLA-B27 is associated with reduced disease activity in axial spondyloarthritis. *Scientific Reports* 2021; 11: 1-5.
5. Pastrello C, Tavasolian F, Ahmed Z et al. Vesicular traffic-mediated cell-to-cell signaling at the immune synapse in Ankylosing Spondylitis. *Frontiers in Immunology* 13: 8070.
6. Mauro D, Thomas R, Guggino G et al. Ankylosing spondylitis: an autoimmune or autoinflammatory disease? *Nature Reviews Rheumatology* 2021; 17: 387-404.
7. Gracey E, Yao Y, Qaiyum Z et al. Altered cytotoxicity profile of CD 8+ T cells in ankylosing spondylitis. *Arthritis & Rheumatology* 2020; 72: 428-434.
8. Cowardin CA, Kuehne SA, Buonomo EL et al. Inflammasome activation contributes to interleukin-23 production in response to *Clostridium difficile*. *MBio* 2015; 6: e02386-02314.
9. Lamkanfi M, Dixit VM. Mechanisms and functions of inflammasomes. *Cell* 2014; 157: 1013-1022.
10. Gracey E, Qaiyum Z, Almaghlouth I et al. IL-7 primes IL-17 in mucosal-associated invariant T (MAIT) cells, which contribute to the Th17-axis in ankylosing spondylitis. *Annals of the rheumatic diseases* 2016; 75: 2124-2132.
11. Qaiyum Z, Lim M, Inman RD. The gut-joint axis in spondyloarthritis: immunological, microbial, and clinical insights. In *Seminars in immunopathology*. Springer 2021; 173-192.
12. Tavasolian F, Inman RD. Gut microbiota–microRNA interactions in ankylosing spondylitis. *Autoimmunity Reviews* 2021; 20: 102827.
13. Lories R, Matthys P, De Vlam K et al. Ankylosing enthesitis, dactylitis, and onychoprosperiostitis in male DBA/1 mice: a model of psoriatic arthritis. *Annals of the rheumatic diseases* 2004; 63: 595-598.
14. Jacques P, Lambrecht S, Verheugen E et al. Proof of concept: enthesitis and new bone formation in spondyloarthritis are driven by mechanical strain and stromal cells. *Annals of the rheumatic diseases* 2014; 73: 437-445.
15. Kusuda M, Haroon N, Nakamura A. Complexity of enthesitis and new bone formation in ankylosing spondylitis: current understanding of the immunopathology and therapeutic approaches. *Modern Rheumatology* 2022; 32: 484-492.
16. Shen H, Wang S, Chen F et al. RNA Sequencing Reveals the Expression Profiles of circRNAs and Indicates Hsa_circ_0070562 as a Pro-osteogenic Factor in Bone Marrow-Derived Mesenchymal Stem Cells of Patients With Ankylosing Spondylitis. *Frontiers in Genetics* 2022; 1694.
17. Papagoras C, Chrysanthopoulou A, Mitsios A et al. IL-17A expressed on neutrophil extracellular traps promotes mesenchymal stem cell differentiation toward bone-forming cells in ankylosing spondylitis. *European journal of immunology* 2021; 51: 930-942.
18. Xie Z, Wang P, Li Y et al. Imbalance between bone morphogenetic protein 2 and noggin induces abnormal osteogenic differentiation of mesenchymal stem cells in ankylosing spondylitis. *Arthritis & Rheumatology* 2016; 68: 430-440.

19. Xie Z, Wang P, Li J et al. MCP1 triggers monocyte dysfunctions during abnormal osteogenic differentiation of mesenchymal stem cells in ankylosing spondylitis. *Journal of Molecular Medicine* 2017; 95: 143-154.
20. Baraniak PR, McDevitt TC. Stem cell paracrine actions and tissue regeneration. *Regenerative medicine* 2010; 5: 121-143.
21. Don W. First off-the-shelf mesenchymal stem cell therapy nears European approval. *Nature Biotechnology* 2018; 36: 213.
22. Mackay AM, Beck SC, Murphy JM et al. Chondrogenic differentiation of cultured human mesenchymal stem cells from marrow. *Tissue engineering* 1998; 4: 415-428.
23. Pittenger MF, Mackay AM, Beck SC et al. Multilineage potential of adult human mesenchymal stem cells. *science* 1999; 284: 143-147.
24. Crippa S, Bernardo ME. Mesenchymal stromal cells: role in the BM niche and in the support of hematopoietic stem cell transplantation. *Hemasphere* 2018; 2.
25. Wei Q, Frenette PS. Niches for hematopoietic stem cells and their progeny. *Immunity* 2018; 48: 632-648.
26. Asada N, Takeishi S, Frenette PS. Complexity of bone marrow hematopoietic stem cell niche. *International journal of hematology* 2017; 106: 45-54.
27. Wang Y, Chen X, Cao W, Shi Y. Plasticity of mesenchymal stem cells in immunomodulation: pathological and therapeutic implications. *Nature immunology* 2014; 15: 1009-1016.
28. Dominici M, Le Blanc K, Mueller I et al. Minimal criteria for defining multipotent mesenchymal stromal cells. The International Society for Cellular Therapy position statement. *Cytotherapy* 2006; 8: 315-317.
29. Gao F, Chiu S, Motan D et al. Mesenchymal stem cells and immunomodulation: current status and future prospects. *Cell death & disease* 2016; 7: e2062-e2062.
30. Huaman O, Bahamonde J, Cahuascanco B et al. Immunomodulatory and immunogenic properties of mesenchymal stem cells derived from bovine fetal bone marrow and adipose tissue. *Research in veterinary science* 2019; 124: 212-222.
31. Berglund AK, Fortier LA, Antczak DF, Schnabel LV. Immunoprivileged no more: measuring the immunogenicity of allogeneic adult mesenchymal stem cells. *Stem cell research & therapy* 2017; 8: 1-7.
32. Wang Y, Tian M, Wang F et al. Understanding the immunological mechanisms of mesenchymal stem cells in allogeneic transplantation: from the aspect of major histocompatibility complex class I. *Stem cells and development* 2019; 28: 1141-1150.
33. Jiang W, Xu J. Immune modulation by mesenchymal stem cells. *Cell proliferation* 2020; 53: e12712.
34. Novoseletskaia E, Grigorieva O, Nimiritsky P et al. Mesenchymal stromal cell-produced components of extracellular matrix potentiate multipotent stem cell response to differentiation stimuli. *Frontiers in Cell and Developmental Biology* 2020; 8: 555378.
35. Boehm T. Evolution of vertebrate immunity. *Current Biology* 2012; 22: R722-R732.
36. Reis J, Vender R, Torres T. Bimekizumab: the first dual inhibitor of interleukin (IL)-17A and IL-17F for the treatment of psoriatic disease and ankylosing spondylitis. *BioDrugs* 2019; 33: 391-399.
37. Kular J, Tickner J, Chim SM, Xu J. An overview of the regulation of bone remodelling at the cellular level. *Clinical biochemistry* 2012; 45: 863-873.
38. Karsenty G, Kronenberg HM, Settembre C. Genetic control of bone formation. *Annual Review of Cell and Developmental* 2009; 25: 629-648.
39. Adamopoulos IE. Inflammation in bone physiology and pathology. *Current opinion in rheumatology* 2018; 30: 59.
40. Rudwaleit M, Van Der Heijde D, Landewé R et al. The development of Assessment of SpondyloArthritis international Society classification criteria for axial spondyloarthritis (part II): validation and final selection. *Annals of the rheumatic diseases* 2009; 68: 777-783.

41. Sawicki LM, Lütje S, Baraliakos X et al. Dual-phase hybrid 18F-Fluoride Positron emission tomography/MRI in ankylosing spondylitis: Investigating the link between MRI bone changes, regional hyperaemia and increased osteoblastic activity. *Journal of Medical Imaging and Radiation Oncology* 2018; 62: 313-319.
42. Watad A, Bridgwood C, Russell T et al. The early phases of ankylosing spondylitis: emerging insights from clinical and basic science. *Frontiers in immunology* 2018; 9: 2668.
43. Jung J-Y, Han SH, Hong YS et al. Inflammation on spinal magnetic resonance imaging is associated with poor bone quality in patients with ankylosing spondylitis. *Modern Rheumatology* 2019; 29: 829-835.
44. Liu L, Yuan Y, Zhang S et al. Osteoimmunological insights into the pathogenesis of ankylosing spondylitis. *Journal of Cellular Physiology* 2021; 236: 6090-6100.
45. Bernardo ME, Fibbe WE. Mesenchymal stromal cells: sensors and switchers of inflammation. *Cell stem cell* 2013; 13: 392-402.
46. Uccelli A, Moretta L, Pistoia V. Mesenchymal stem cells in health and disease. *Nature reviews immunology* 2008; 8: 726-736.
47. Stewart MC, Stewart AA. Mesenchymal stem cells: characteristics, sources, and mechanisms of action. *Veterinary Clinics: Equine Practice* 2011; 27: 243-261.
48. Lories RJ, Luyten FP, De Vlam K. Progress in spondylarthritis. Mechanisms of new bone formation in spondyloarthritis. *Arthritis research & therapy* 2009; 11: 1-8.
49. Cao X. RANKL-RANK signaling regulates osteoblast differentiation and bone formation. *Bone research* 2018; 6: 1-2.
50. Uehara T, Mise-Omata S, Matsui M et al. Delivery of RANKL-Binding Peptide OP3-4 Promotes BMP-2-Induced Maxillary Bone Regeneration. *Journal of Dental Research* 2016; 95: 665-672.
51. Portal-Núñez S, Mediero A, Esbrit P et al. Unexpected bone formation produced by RANKL blockade. *Trends in Endocrinology & Metabolism* 2017; 28: 695-704.
52. Ikebuchi Y, Aoki S, Honma M et al. Coupling of bone resorption and formation by RANKL reverse signalling. *Nature* 2018; 561: 195-200.
53. Wu Y, Ren M, Yang R et al. Reduced immunomodulation potential of bone marrow-derived mesenchymal stem cells induced CCR4+ CCR6+ Th/Treg cell subset imbalance in ankylosing spondylitis. *Arthritis research & therapy* 2011; 13: 1-15.
54. Okamoto K. Regulation of bone by IL-17-producing T cells. *Nihon Rinsho Men'eki Gakkai Kaishi= Japanese Journal of Clinical Immunology* 2017; 40: 361-366.
55. Daoussis D, Kanellou A, Panagiotopoulos E, Papachristou D. DKK-1 Is Underexpressed in Mesenchymal Stem Cells from Patients with Ankylosing Spondylitis and Further Downregulated by IL-17. *International journal of molecular sciences* 2022; 23: 6660.
56. He T, Huang Y, Zhang C et al. Interleukin-17A-promoted MSC2 polarization related with new bone formation of ankylosing spondylitis. *Oncotarget* 2017; 8: 96993.
57. Akiyama K, Chen C, Wang D et al. Mesenchymal-stem-cell-induced immunoregulation involves FAS-ligand-/FAS-mediated T cell apoptosis. *Cell stem cell* 2012; 10: 544-555.
58. Ren G, Zhang L, Zhao X et al. Mesenchymal stem cell-mediated immunosuppression occurs via concerted action of chemokines and nitric oxide. *Cell stem cell* 2008; 2: 141-150.
59. Li D, Wang P, Li Y et al. All-trans retinoic acid improves the effects of bone marrow-derived mesenchymal stem cells on the treatment of ankylosing spondylitis: an in vitro study. *Stem cells international* 2015; 2015.
60. Xie Z, Yu W, Zheng G et al. TNF- α -mediated m6A modification of ELMO1 triggers directional migration of mesenchymal stem cell in ankylosing spondylitis. *Nature communications* 2021; 12: 1-14.
61. De Bari C. Are mesenchymal stem cells in rheumatoid arthritis the good or bad guys? *Arthritis research & therapy* 2015; 17: 1-9.

62. Abdolmohammadi K, Pakdel FD, Aghaei H et al. Ankylosing spondylitis and mesenchymal stromal/stem cell therapy: a new therapeutic approach. *Biomedicine & Pharmacotherapy* 2019; 109: 1196-1205.
63. Santos JM, Bárçia RN, Simões SI et al. The role of human umbilical cord tissue-derived mesenchymal stromal cells (UCX®) in the treatment of inflammatory arthritis. *Journal of translational medicine* 2013; 11: 1-13.
64. Oppmann B, Lesley R, Blom B et al. Novel p19 protein engages IL-12p40 to form a cytokine, IL-23, with biological activities similar as well as distinct from IL-12. *Immunity* 2000; 13: 715-725.
65. Gaffen SL, Jain R, Garg AV, Cua DJ. The IL-23–IL-17 immune axis: from mechanisms to therapeutic testing. *Nature reviews immunology* 2014; 14: 585-600.
66. Razawy W, van Driel M, Lubberts E. The role of IL-23 receptor signaling in inflammation-mediated erosive autoimmune arthritis and bone remodeling. *European Journal of Immunology* 2018; 48: 220-229.
67. Sato K, Suematsu A, Okamoto K et al. Th17 functions as an osteoclastogenic helper T cell subset that links T cell activation and bone destruction. *The Journal of experimental medicine* 2006; 203: 2673-2682.
68. Evans DM, Spencer CC, Pointon JJ et al. Interaction between ERAP1 and HLA-B27 in ankylosing spondylitis implicates peptide handling in the mechanism for HLA-B27 in disease susceptibility. *Nature genetics* 2011; 43: 761-767.
69. Nakamura A, Zeng F, Nakamura S et al. Macrophage migration inhibitory factor drives pathology in a mouse model of spondyloarthritis and is associated with human disease. *Science Translational Medicine* 2021; 13: eabg1210.
70. Pittenger MF, Discher DE, Péault BM et al. Mesenchymal stem cell perspective: cell biology to clinical progress. *NPJ Regenerative medicine* 2019; 4: 1-15.
71. Pedersen SJ, Maksymowych WP. The pathogenesis of ankylosing spondylitis: an update. *Current Rheumatology Reports* 2019; 21: 1-10.
72. Hammitzsch A, Chen L, de Wit J et al. Inhibiting ex-vivo Th17 responses in ankylosing spondylitis by targeting Janus kinases. *Scientific reports* 2018; 8: 1-8.
73. Ankrum JA, Ong JF, Karp JM. Mesenchymal stem cells: immune evasive, not immune privileged. *Nature biotechnology* 2014; 32: 252-260.
74. Wang P, Li Y, Huang L et al. Effects and safety of allogenic mesenchymal stem cell intravenous infusion in active ankylosing spondylitis patients who failed NSAIDs: a 20-week clinical trial. *Cell transplantation* 2014; 23: 1293-1303.
75. Zeng L, Yu G, Yang K et al. Efficacy and Safety of Mesenchymal Stem Cell Transplantation in the Treatment of Autoimmune Diseases (Rheumatoid Arthritis, Systemic Lupus Erythematosus, Inflammatory Bowel Disease, Multiple Sclerosis, and Ankylosing Spondylitis): A Systematic Review and Meta-Analysis of Randomized Controlled Trial. *Stem cells international* 2022; 2022.
76. Fang W, Zhang Y, Chen Z. Innate lymphoid cells in inflammatory arthritis. *Arthritis Research & Therapy* 2020; 22: 1-7.
77. Bennisstein SB, Weinhold S, Degistirici Ö et al. Efficient in vitro generation of IL-22-secreting ILC3 from CD34+ hematopoietic progenitors in a human mesenchymal stem cell niche. *Frontiers in immunology* 2021; 12.
78. Ye G, Xie Z, Zeng H et al. Oxidative stress-mediated mitochondrial dysfunction facilitates mesenchymal stem cell senescence in ankylosing spondylitis. *Cell death & disease* 2020; 11: 1-13.
79. Gilkeson GS. Safety and Efficacy of Mesenchymal Stromal Cells and Other Cellular Therapeutics in Rheumatic Diseases in 2022: A review of what we know so far. *Arthritis & Rheumatology* 2022; 74: 752-765.
80. Bianco P, Cao X, Frenette PS et al. The meaning, the sense and the significance: translating the science of mesenchymal stem cells into medicine. *Nature medicine* 2013; 19: 35-42.
81. Schrepfer S, Deuse T, Reichenspurner H et al. Stem cell transplantation: the lung barrier. In *Transplantation proceedings*. Elsevier 2007; 573-576.

82. Tyndall A. Mesenchymal stem cell treatments in rheumatology—a glass half full? *Nature Reviews Rheumatology* 2014; 10: 117-124.
83. Chilima TDP, Moncaubeig F, Farid SS. Impact of allogeneic stem cell manufacturing decisions on cost of goods, process robustness and reimbursement. *Biochemical Engineering Journal* 2018; 137: 132-151.
84. Levy O, Kuai R, Siren EM et al. Shattering barriers toward clinically meaningful MSC therapies. *Science Advances* 2020; 6: eaba6884.
85. Shammaa R, El-Kadiry AE-H, Abusarah J, Rafei M. Mesenchymal stem cells beyond regenerative medicine. *Frontiers in cell and developmental biology* 2020; 8: 72.
86. Wright A, Arthaud-Day ML, Weiss ML. Therapeutic use of mesenchymal stromal cells: the need for inclusive characterization guidelines to accommodate all tissue sources and species. *Frontiers in Cell and Developmental Biology* 2021; 9: 632717.
87. Gao L, McBeath R, Chen CS. Stem cell shape regulates a chondrogenic versus myogenic fate through Rac1 and N-cadherin. *Stem cells* 2010; 28: 564-572.
88. Kilian KA, Bugarija B, Lahn BT, Mrksich M. Geometric cues for directing the differentiation of mesenchymal stem cells. *Proceedings of the National Academy of Sciences* 2010; 107: 4872-4877.
89. McBeath R, Pirone DM, Nelson CM et al. Cell shape, cytoskeletal tension, and RhoA regulate stem cell lineage commitment. *Developmental cell* 2004; 6: 483-495.
90. Khetan S, Guvendiren M, Legant WR et al. Degradation-mediated cellular traction directs stem cell fate in covalently crosslinked three-dimensional hydrogels. *Nature materials* 2013; 12: 458-465.
91. Huebsch N, Arany PR, Mao AS et al. Harnessing traction-mediated manipulation of the cell/matrix interface to control stem-cell fate. *Nature materials* 2010; 9: 518-526.
92. Lo Surdo J, Bauer SR. Quantitative approaches to detect donor and passage differences in adipogenic potential and clonogenicity in human bone marrow-derived mesenchymal stem cells. *Tissue engineering part C: methods* 2012; 18: 877-889.
93. Zaim M, Karaman S, Cetin G, Isik S. Donor age and long-term culture affect differentiation and proliferation of human bone marrow mesenchymal stem cells. *Annals of hematology* 2012; 91: 1175-1186.
94. Vega SL, Liu E, Patel PJ et al. High-content imaging-based screening of microenvironment-induced changes to stem cells. *Journal of biomolecular screening* 2012; 17: 1151-1162.
95. Marklein RA, Lo Surdo JL, Bellayr IH et al. High content imaging of early morphological signatures predicts long term mineralization capacity of human mesenchymal stem cells upon osteogenic induction. *Stem Cells* 2016; 34: 935-947.
96. Lötvall J, Hill AF, Hochberg F et al. Minimal experimental requirements for definition of extracellular vesicles and their functions: a position statement from the International Society for Extracellular Vesicles. In. *Taylor & Francis* 2014; 26913.
97. Raposo G, Stoorvogel W. Extracellular vesicles: exosomes, microvesicles, and friends. *Journal of Cell Biology* 2013; 200: 373-383.
98. Das S, Abdel-Mageed AB, Adamidi C et al. The extracellular RNA communication consortium: establishing foundational knowledge and technologies for extracellular RNA research. *Cell* 2019; 177: 231-242.
99. Théry C, Witwer KW, Aikawa E et al. Minimal information for studies of extracellular vesicles 2018 (MISEV2018): a position statement of the International Society for Extracellular Vesicles and update of the MISEV2014 guidelines. *Journal of extracellular vesicles* 2018; 7: 1535750.
100. Colombo M, Raposo G, Théry C. Biogenesis, secretion, and intercellular interactions of exosomes and other extracellular vesicles. *Annual review of cell and developmental biology* 2014; 30: 255-289.
101. Nguyen HP, Simpson RJ, Salamonsen LA, Greening DW. Extracellular vesicles in the intrauterine environment: challenges and potential functions. *Biology of Reproduction* 2016; 95: 109, 101-112.

102. Van Niel G, d'Angelo G, Raposo G. Shedding light on the cell biology of extracellular vesicles. *Nature reviews Molecular cell biology* 2018; 19: 213-228.
103. van Niel G, Carter DR, Clayton A et al. Challenges and directions in studying cell–cell communication by extracellular vesicles. *Nature Reviews Molecular Cell Biology* 2022; 23: 369-382.
104. O'Brien K, Breynne K, Ughetto S et al. RNA delivery by extracellular vesicles in mammalian cells and its applications. *Nature reviews Molecular cell biology* 2020; 21: 585-606.
105. Huang Y, Feng F, Huang Q et al. Proteomic analysis of serum-derived extracellular vesicles in ankylosing spondylitis patients. *International Immunopharmacology* 2020; 87: 106773.
106. Huang Y, Liu Y, Huang Q et al. TMT-Based Quantitative Proteomics Analysis of Synovial Fluid-Derived Exosomes in Inflammatory Arthritis. *Frontiers in immunology* 2022; 13: 800902-800902.
107. Shirazi S, Huang C-C, Kang M et al. The importance of cellular and exosomal miRNAs in mesenchymal stem cell osteoblastic differentiation. *Scientific reports* 2021; 11: 1-14.
108. Meulenbelt I, Ramos YF, Baglio SR, Pegtel DM. Censoring exosomal crosstalk in osteoarthritis. *Nature Aging* 2021; 1: 332-334.
109. Willms E, Cabañas C, Mäger I et al. Extracellular vesicle heterogeneity: subpopulations, isolation techniques, and diverse functions in cancer progression. *Frontiers in immunology* 2018; 9: 738.
110. Tang K, Zhang Y, Zhang H et al. Delivery of chemotherapeutic drugs in tumour cell-derived microparticles. *Nature communications* 2012; 3: 1-11.
111. Herrmann IK, Wood MJA, Fuhrmann G. Extracellular vesicles as a next-generation drug delivery platform. *Nature nanotechnology* 2021; 16: 748-759.
112. Arabpour M, Saghadzadeh A, Rezaei N. Anti-inflammatory and M2 macrophage polarization-promoting effect of mesenchymal stem cell-derived exosomes. *International Immunopharmacology* 2021; 97: 107823.
113. Zhao H, Shang Q, Pan Z et al. Exosomes from adipose-derived stem cells attenuate adipose inflammation and obesity through polarizing M2 macrophages and beiging in white adipose tissue. *Diabetes* 2018; 67: 235-247.
114. Heo JS, Choi Y, Kim HO. Adipose-derived mesenchymal stem cells promote M2 macrophage phenotype through exosomes. *Stem cells international* 2019; 2019.
115. Wang J, Huang R, Xu Q et al. Mesenchymal Stem Cell–Derived Extracellular Vesicles Alleviate Acute Lung Injury Via Transfer of miR-27a-3p. *Critical Care Medicine* 2020; 48: e599-e610.
116. Harrell CR, Jovicic N, Djonov V et al. Mesenchymal stem cell-derived exosomes and other extracellular vesicles as new remedies in the therapy of inflammatory diseases. *Cells* 2019; 8: 1605.
117. Sohrabi B, Dayeri B, Zahedi E et al. Mesenchymal stem cell (MSC)-derived exosomes as novel vehicles for delivery of miRNAs in cancer therapy. *Cancer Gene Therapy* 2022; 1-12.
118. Fang S, Liu Z, Wu S et al. Pro-angiogenic and pro-osteogenic effects of human umbilical cord mesenchymal stem cell-derived exosomal miR-21-5p in osteonecrosis of the femoral head. *Cell Death Discovery* 2022; 8: 1-11.
119. Zhu D, Tian J, Wu X et al. G-MDSC-derived exosomes attenuate collagen-induced arthritis by impairing Th1 and Th17 cell responses. *Biochimica et Biophysica Acta (BBA)-Molecular Basis of Disease* 2019; 1865: 165540.
120. Mohanty A, Poliseti N, Vemuganti GK. Immunomodulatory properties of bone marrow mesenchymal stem cells. *Journal of Biosciences* 2020; 45: 1-17.
121. Cosenza S, Toupet K, Maumus M et al. Mesenchymal stem cells-derived exosomes are more immunosuppressive than microparticles in inflammatory arthritis. *Theranostics* 2018; 8: 1399.
122. Li Y-J, Chen Z. Cell-based therapies for rheumatoid arthritis: opportunities and challenges. *Therapeutic Advances in Musculoskeletal Disease* 2022; 14: 1759720X221100294.
123. Kou M, Huang L, Yang J et al. Mesenchymal stem cell-derived extracellular vesicles for immunomodulation and regeneration: a next generation therapeutic tool? *Cell Death & Disease* 2022; 13: 1-16.

124. Kobayashi H, Singer RH. Single-molecule imaging of microRNA-mediated gene silencing in cells. *Nature communications* 2022; 13: 1-14.
125. Kim S, Kim S, Chang HR et al. The regulatory impact of RNA-binding proteins on microRNA targeting. *Nature communications* 2021; 12: 1-15.
126. Neviani P, Wise PM, Murtadha M et al. Natural killer–derived exosomal miR-186 inhibits neuroblastoma growth and immune escape mechanisms. *Cancer research* 2019; 79: 1151-1164.
127. Batista IA, Quintas ST, Melo SA. The interplay of exosomes and NK cells in cancer biology. *Cancers* 2021; 13: 473.
128. Garcia-Martin R, Wang G, Brandão BB et al. MicroRNA sequence codes for small extracellular vesicle release and cellular retention. *Nature* 2022; 601: 446-451.
129. Liu J, Wu X, Lu J et al. Exosomal transfer of osteoclast-derived miRNAs to chondrocytes contributes to osteoarthritis progression. *Nature Aging* 2021; 1: 368-384.
130. Liu W, Wang P, Xie Z et al. Abnormal inhibition of osteoclastogenesis by mesenchymal stem cells through the miR-4284/CXCL5 axis in ankylosing spondylitis. *Cell death & disease* 2019; 10: 1-11.
131. Liu C, Liang T, Zhang Z et al. Transfer of microRNA-22-3p by M2 macrophage-derived extracellular vesicles facilitates the development of ankylosing spondylitis through the PER2-mediated Wnt/ β -catenin axis. *Cell death discovery* 2022; 8: 1-12.
132. Baek G, Choi H, Kim Y et al. Mesenchymal stem cell-derived extracellular vesicles as therapeutics and as a drug delivery platform. *Stem cells translational medicine* 2019; 8: 880-886.
133. Zhao J, Zhang Y, Liu B. MicroRNA-204-5p inhibits the osteogenic differentiation of ankylosing spondylitis fibroblasts by regulating the Notch2 signaling pathway. *Molecular Medicine Reports* 2020; 22: 2537-2544.
134. Du W, Yin L, Tong P et al. MiR-495 targeting dvl-2 represses the inflammatory response of ankylosing spondylitis. *American Journal of Translational Research* 2019; 11: 2742.
135. Qin X, Zhu B, Jiang T et al. miR-17-5p regulates heterotopic ossification by targeting ANKH in ankylosing spondylitis. *Molecular Therapy-Nucleic Acids* 2019; 18: 696-707.
136. Hu L, Guan Z, Tang C et al. Exosomes derived from microRNA-21 overexpressed adipose tissue-derived mesenchymal stem cells alleviate spine osteoporosis in ankylosing spondylitis mice. *Journal of Tissue Engineering and Regenerative Medicine* 2022.
137. Han Y, Yang J, Fang J et al. The secretion profile of mesenchymal stem cells and potential applications in treating human diseases. *Signal Transduction and Targeted Therapy* 2022; 7: 1-19.
138. Mohammadi H, Hemmatzadeh M, Babaie F et al. MicroRNA implications in the etiopathogenesis of ankylosing spondylitis. *Journal of cellular physiology* 2018; 233: 5564-5573.
139. Bonsergent E, Grisard E, Buchrieser J et al. Quantitative characterization of extracellular vesicle uptake and content delivery within mammalian cells. *Nature communications* 2021; 12: 1-11.
140. O'Brien K, Ughetto S, Mahjoum S et al. Uptake, functionality, and re-release of extracellular vesicle-encapsulated cargo. *Cell reports* 2022; 39: 110651.
141. Joshi BS, de Beer MA, Giepmans BN, Zuhorn IS. Endocytosis of extracellular vesicles and release of their cargo from endosomes. *ACS nano* 2020; 14: 4444-4455.

Response To Reviewers Letter

Dear Reviewers:

Thank you for the constructive comments concerning our manuscript entitled “Mesenchymal stem cell extracellular vesicles: Biology and therapeutic potential in Axial Spondyloarthritis”.

These comments were valuable and very helpful for improving our manuscript to better demonstrate the important significance of our research. We have carefully reviewed all comments and completed point-by-point revisions. The responses to the comments in this letter and the revised portions of the manuscript are marked in colors. We tried to delete/rewrite all the unclear sentences or the ones that have a lot of contradictory and uncertainty about them in the literature. All the missing references are added to the text. We appreciate the work of the Reviewers and hope that the revisions will meet with approval. We will be glad to respond to any further comments that you may have.

Yours sincerely,

Dr. Robert D. Inman

Schroeder Arthritis Institute, Toronto Western Hospital, 399 Bathurst Street, Room1E-423, Toronto, Ontario M5T 2S8, Canada.

Robert.Inman@uhn.ca

Reviewer #1 (Remarks to the Author):

Response: Thank you for your constructive comments on the manuscript. This time, we attempted to describe the cell biology of MSCs and their therapeutic progress, assess the several roles of MSCs in the development and treatment of AxSpA, and offer EVs-derived MSCs as a potential new therapy for AxSpA. We begin with the cell biology of MSCs and their therapeutic progress, then describe the bone remodeling process in AxSpA. Then, we show evidence that MSCs exhibit increased osteoblast differentiation in AxSpA. Then we define the approach using external MSCs in AxSpA following clinical trials of MSCs in AxSpA. Following that, we discussed the limitations of MSC-based therapy for AxSpA. Then, we offered MSC-derived EVs as a new cell-free therapy and defined treatments based on MSC-derived EVs, after providing some data about the challenges and uses of EVS generated from MSCs as an innovative treatment for AxSpA.

Reviewers' comments:

Reviewer #1 (Remarks to the Author):

This second revision, COMMSBIO-22-2597B still does not capture the latest scientific developments in the field. While the change in title now justifies the dominance of MSCs over MSC EVs in the review, the review of MSC EVs is still inadequate, confusing and even wrong. For example, the sentence in the abstract "Compounds contained within extracellular vehicles (EVs) generated by MSCs have been proven to reproduce the impact of MSCs on target cells." is wrong. There is no definitive report to date that demonstrates compounds in EVs reproduce the impact of MSCs on target cells. A scientifically accurate statement would be "extracellular vehicles (EVs) generated by MSCs have been proven to reproduce the impact of MSCs on target cells." The statement "EVs biological effects rely on their cargo, with microRNAs (miRNAs) integrated into EVs playing a particularly important role in gene expression regulation." is incongruent with their statement on p10 "However, recent observations on the low efficiency of EVs absorption by cells, the escape of EVs from the endosome/lysosome pathway, and the uncoating of EVs to release their contents in the cytosol have rendered the function of miRNA in regulating the activity of MSC- derived EVs more questionable.[139-141]."

As I mentioned earlier, many issues regarding MSC biology and therapeutic activities have already been extensively covered in many reviews. These issues are relatively generic and applicable to all diseases including Axial Spondyloarthritis. A potential redeeming feature for this review would be the contemporary review of MSC EVs in the context of Axial Spondyloarthritis. Unfortunately, many of the key seminal papers on the effects of MSC EVs on cartilage and bone were not included. Attached is a list papers from a simple pubmed search using as search terms, "MSC, exosomes, extracellular vesicles, bone, cartilage". Many of these including the seminal one were not even mentioned. Similarly, many of the seminal papers on the immune activities of MSC exosomes were also not included. While it is not reasonable to expect the authors to conduct an exhaustive review of all the papers in the field, key papers should be cited and reviewed. In addition, the discussion on the miRNA cargo in EVs without including other papers demonstrating the role of EV proteins and not miRNA in mediating the effects of MSC EVs is not scientifically objective. Also , ending the long discourse with "However, recent observations on the low efficiency of EVs absorption by cells, the escape of EVs from the endosome/lysosome pathway, and the uncoating of EVs to release their contents in the cytosol have rendered the function of miRNA in regulating the activity of MSC- derived EVs more questionable.[139-141]." simply negates the discourse and questions the utility of having the discourse in the first place.

As a review of a relatively new field, it would be useful to provide a narrative of the iterative development of the various aspects from first discoveries which in this case are generally a couple of years old to the present state. This narrative should also be balanced. It should not be biased as exemplified by the section on the miRNA.

Response To Reviewers Letter

Dear Reviewers:

Thank you for the constructive comments concerning our manuscript entitled “Mesenchymal stem cell extracellular vesicles: Biology and therapeutic potential in Axial Spondyloarthritis”.

These comments were valuable and very helpful for improving our manuscript to better demonstrate the important significance of our research. We have carefully reviewed all comments and completed point-by-point revisions. The responses to the comments in this letter and the revised portions of the manuscript are marked in colors. We tried to delete/rewrite all the unclear sentences or the ones that have a lot of contradictory and uncertainty about them in the literature. All the missing references are added to the text. We appreciate the work of the Reviewers and hope that the revisions will meet with approval. We will be glad to respond to any further comments that you may have.

Yours sincerely,

Dr. Robert D. Inman

Schroeder Arthritis Institute, Toronto Western Hospital, 399 Bathurst Street, Room1E-423, Toronto, Ontario M5T 2S8, Canada.

Robert.Inman@uhn.ca

Reviewer #1 (Remarks to the Author):

This second revision, COMMSBIO-22-2597B still does not capture the latest scientific developments in the field. While the change in the title now justifies the dominance of MSCs over MSC EVs in the review, the review of MSC EVs is still inadequate, confusing and even wrong. For example, the sentence in the abstract “Compounds contained within extracellular vehicles (EVs) generated by MSCs have been proven to reproduce the impact of MSCs on target cells.” is wrong. There is no definitive report to date that demonstrates compounds in EVs reproduce the impact of MSCs on target cells. A scientifically accurate statement would be “extracellular vehicles (EVs) generated by MSCs have been proven to reproduce the impact of MSCs on target cells.”

We have made the change in this statement in accordance with the reviewer’s suggestion.

The statement “EVs biological effects rely on their cargo, with microRNAs (miRNAs) integrated into EVs playing a particularly important role in gene expression regulation.” is incongruent with their statement on p10 “However, recent observations on the low efficiency of EVs absorption by cells, the escape of EVs from the endosome/lysosome pathway, and the uncoating of EVs to release their contents in the cytosol have rendered the function of miRNA in regulating the activity of MSC-derived EVs more questionable”. [139-141].

We have clarified this and thank the reviewer in this regard.

As I mentioned earlier, many issues regarding MSC biology and therapeutic activities have already been extensively covered in many reviews. These issues are relatively generic and applicable to all diseases including Axial Spondyloarthritis. A potential redeeming feature for this review would be the contemporary review of MSC EVs in the context of Axial Spondyloarthritis. Unfortunately, many of the key seminal papers on the effects of MSC EVs on cartilage and bone were not included. Attached is a list papers from a simple pubmed search using as search terms, “MSC, exosomes, extracellular vesicles, bone, cartilage”. Many of these including the seminal one were not even mentioned. Similarly, many of the seminal papers on the immune activities of MSC exosomes were also not included. While it is not reasonable to expect the authors to conduct an exhaustive review of all the papers in the field, key papers should be cited and reviewed.

We thank the reviewer for these relevant references and they have been cited and included in the discussion.

In addition, the discussion on the miRNA cargo in EVs without including other papers demonstrating the role of EV proteins and not miRNA in mediating the effects of MSC EVs is not scientifically objective. Also, ending the long discourse with “However, recent observations on the low efficiency of EVs absorption by cells, the escape of EVs from the endosome/lysosome pathway, and the uncoating of EVs to release their contents in the cytosol have rendered the function of miRNA in regulating the activity of MSC- derived EVs more questionable.[139-141].” simply negates the discourse and questions the utility of having the discourse in the first place.

The reviewer makes an insightful comment once again. We have moderated the “questionable” comments since we agree they seem to undermine the review.

As a review of a relatively new field, it would be useful to provide a narrative of the iterative development of the various aspects from first discoveries which in this case are generally a couple of years old to the present state. This narrative should also be balanced. It should not be biased as exemplified by the section on the miRNA.

Response: Thank you for your constructive comments on the manuscript.

We tried to start we brief introduction about the pathophysiology of AxSpA. MSCs' point of view and their biology to therapeutic advancement in AxSpA followed by the role of MSCs-derived EVs in AxSpA. In the introduction, we emphasize more the role of the immune system and inflammation which causes new bone formation in AxSpA. The MSC section explains the immunomodulatory effect of MSCs and their dual role in AxSpA. and finally, we mentioned some clinical trials of MSCs in AxSpA as a shred of evidence. In the EV section, we tried to emphasize the role of EVs on bone and cartilage regeneration and mentioned the role of articular chondrocyte-derived exosomes and synoviocytes-derived exosomes followed by explaining the role of EVs in AxSpA. Then, we offered MSC-derived EVs as a new cell-free therapy and defined treatments based on MSC-derived EVs, after providing some data about the challenges and uses of EVs generated from MSCs as an innovative treatment for AxSpA.

Response To Reviewers Letter

Dear Reviewers:

Thank you for the constructive comments concerning our manuscript entitled “Mesenchymal stem cell extracellular vesicles: Biology and therapeutic potential in Axial Spondyloarthritis”.

These comments were valuable and very helpful for improving our manuscript to better demonstrate the important significance of our research. We have carefully reviewed all comments and completed point-by-point revisions. The responses to the comments in this letter and the revised portions of the manuscript are marked in colors. We tried to delete/rewrite all the unclear sentences or the ones that have a lot of contradictory and uncertainty about them in the literature. All the missing references are added to the text. We appreciate the work of the Reviewers and hope that the revisions will meet with approval. We will be glad to respond to any further comments that you may have.

Yours sincerely,

Dr. Robert D. Inman

Schroeder Arthritis Institute, Toronto Western Hospital, 399 Bathurst Street, Room1E-423, Toronto, Ontario M5T 2S8, Canada.

Robert.Inman@uhn.ca

Reviewer #1 (Remarks to the Author):

This second revision, COMMSBIO-22-2597B still does not capture the latest scientific developments in the field. While the change in the title now justifies the dominance of MSCs over MSC EVs in the review, the review of MSC EVs is still inadequate, confusing and even wrong. For example, the sentence in the abstract “Compounds contained within extracellular vehicles (EVs) generated by MSCs have been proven to reproduce the impact of MSCs on target cells.” is wrong. There is no definitive report to date that demonstrates compounds in EVs reproduce the impact of MSCs on target cells. A scientifically accurate statement would be “extracellular vehicles (EVs) generated by MSCs have been proven to reproduce the impact of MSCs on target cells.”

We have made the change in this statement in accordance with the reviewer’s suggestion.

The statement “EVs biological effects rely on their cargo, with microRNAs (miRNAs) integrated into EVs playing a particularly important role in gene expression regulation.” is incongruent with their statement on p10 “However, recent observations on the low efficiency of EVs absorption by cells, the escape of EVs from the endosome/lysosome pathway, and the uncoating of EVs to release their contents in the cytosol have rendered the function of miRNA in regulating the activity of MSC-derived EVs more questionable”. [139-141].

We have clarified this and thank the reviewer in this regard.

As I mentioned earlier, many issues regarding MSC biology and therapeutic activities have already been extensively covered in many reviews. These issues are relatively generic and applicable to all diseases including Axial Spondyloarthritis. A potential redeeming feature for this review would be the contemporary review of MSC EVs in the context of Axial Spondyloarthritis. Unfortunately, many of the key seminal papers on the effects of MSC EVs on cartilage and bone were not included. Attached is a list papers from a simple pubmed search using as search terms, “MSC, exosomes, extracellular vesicles, bone, cartilage”. Many of these including the seminal one were not even mentioned. Similarly, many of the seminal papers on the immune activities of MSC exosomes were also not included. While it is not reasonable to expect the authors to conduct an exhaustive review of all the papers in the field, key papers should be cited and reviewed.

We thank the reviewer for these relevant references and they have been cited and included in the discussion.

In addition, the discussion on the miRNA cargo in EVs without including other papers demonstrating the role of EV proteins and not miRNA in mediating the effects of MSC EVs is not scientifically objective. Also, ending the long discourse with “However, recent observations on the low efficiency of EVs absorption by cells, the escape of EVs from the endosome/lysosome pathway, and the uncoating of EVs to release their contents in the cytosol have rendered the function of miRNA in regulating the activity of MSC- derived EVs more questionable.[139-141].” simply negates the discourse and questions the utility of having the discourse in the first place.

The reviewer makes an insightful comment once again. We have moderated the “questionable” comments since we agree they seem to undermine the review.

As a review of a relatively new field, it would be useful to provide a narrative of the iterative development of the various aspects from first discoveries which in this case are generally a couple of years old to the present state. This narrative should also be balanced. It should not be biased as exemplified by the section on the miRNA.

Response: Thank you for your constructive comments on the manuscript.

We tried to start we brief introduction about the pathophysiology of AxSpA. MSCs' point of view and their biology to therapeutic advancement in AxSpA followed by the role of MSCs-derived EVs in AxSpA. In the introduction, we emphasize more the role of the immune system and inflammation which causes new bone formation in AxSpA. The MSC section explains the immunomodulatory effect of MSCs and their dual role in AxSpA. and finally, we mentioned some clinical trials of MSCs in AxSpA as a shred of evidence. In the EV section, we tried to emphasize the role of EVs on bone and cartilage regeneration and mentioned the role of articular chondrocyte-derived exosomes and synoviocytes-derived exosomes followed by explaining the role of EVs in AxSpA. Then, we offered MSC-derived EVs as a new cell-free therapy and defined treatments based on MSC-derived EVs, after providing some data about the challenges and uses of EVs generated from MSCs as an innovative treatment for AxSpA.